# Deep Visual Proteomics maps proteotoxicity in a genetic liver disease

Florian A. Rosenberger[1✉], Sophia C. Mädler[1,13], Katrine Holtz Thorhauge[2,3,13], Sophia Steigerwald[1,13], Malin Fromme[4], Mikhail Lebedev[1], Caroline A. M. Weiss[1], Marc Oeller[1], Maria Wahle[1], Andreas Metousis[1], Maximilian Zwiebel[1], Niklas A. Schmacke[1,5], Sönke Detlefsen[3,6], Peter Boor[7], Ondřej Fabián[8,9], Soňa Fraňková[10], Aleksander Krag[2,3,11], Pavel Strnad[4] & Matthias Mann[1,12✉]

Protein misfolding diseases, including α1-antitrypsin deficiency (AATD), pose substantial health challenges, with their cellular progression still poorly understood[1–3]. We use spatial proteomics by mass spectrometry and machine learning to map AATD in human liver tissue. Combining Deep Visual Proteomics (DVP) with single-cell analysis[4,5], we probe intact patient biopsies to resolve molecular events during hepatocyte stress in pseudotime across fibrosis stages. We achieve proteome depth of up to 4,300 proteins from one-third of a single cell in formalin-fixed, paraffin-embedded tissue. This dataset reveals a potentially clinically actionable peroxisomal upregulation that precedes the canonical unfolded protein response. Our single-cell proteomics data show α1-antitrypsin accumulation is largely cell-intrinsic, with minimal stress propagation between hepatocytes. We integrated proteomic data with artificial intelligence-guided image-based phenotyping across several disease stages, revealing a late-stage hepatocyte phenotype characterized by globular protein aggregates and distinct proteomic signatures, notably including elevated TNFSF10 (also known as TRAIL) amounts. This phenotype may represent a critical disease progression stage. Our study offers new insights into AATD pathogenesis and introduces a powerful methodology for high-resolution, in situ proteomic analysis of complex tissues. This approach holds potential to unravel molecular mechanisms in various protein misfolding disorders, setting a new standard for understanding disease progression at the single-cell level in human tissue.

Spatial omics technologies are revolutionizing our ability to deconvolute molecular events at single-cell resolution within a tissue context. Whereas much focus has been placed on spatial genomics and transcriptomics, recent advances in multiplexed imaging and proteomics are beginning to shed light on the functional proteomic layer. Mass spectrometry (MS)-based proteomics has made significant strides towards biologically informative single-cell analysis, now enabling quantification of up to 5,000 proteins in cultured cells[6–8]. In the tissue context, we have recently introduced Deep Visual Proteomics (DVP), which integrates staining, artificial intelligence-guided cell segmentation and classification, laser microdissection of single-cell shapes and high-sensitivity MS[4,5]. DVP excels in digital pathology applications with pronounced spatial and visual components, providing simultaneous and deep proteomic characterization at the level of thousands of proteins[9].

We reasoned that these emerging technologies would be ideally suited to elucidate molecular events during the progressive worsening of proteotoxicity as it unfolds in patients. Proteotoxicity, characterized by the accumulation of misfolded and aggregated proteins leading to cell damage, is a hallmark of many diseases, including neurodegenerative pathologies such as Alzheimer's disease and Parkinson's disease[10–12]. The underlying cause of proteotoxicity is a disruption in protein homeostasis, resulting in an imbalance between protein synthesis, folding and clearance mechanisms[3].

To investigate proteotoxicity in a clinically relevant context, we focused on a disorder with unmet clinical need that exemplifies the challenges of protein misfolding and aggregation in a vital organ. The fibrogenic liver disease α1-antitrypsin (AAT) deficiency (AATD) is a genetic disorder caused by autosomal, codominant mutations in the *SERPINA1*

[1]Department of Proteomics and Signal Transduction, Max Planck Institute of Biochemistry, Martinsried, Germany. [2]Department of Gastroenterology and Hepatology, Centre for Liver Research, Odense, Denmark. [3]Department of Clinical Research, Faculty of Health Sciences, University of Southern Denmark, Odense, Denmark. [4]Medical Clinic III, Gastroenterology, Metabolic Diseases and Intensive Care, University Hospital RWTH, AachenHealth Care Provider of the European Reference Network on Rare Liver Disorders (ERN RARE LIVER), Aachen, Germany. [5]Gene Center and Department of Biochemistry, Ludwig-Maximilians-Universität München, Munich, Germany. [6]Department of Pathology, Odense University Hospital, Odense, Denmark. [7]Institute of Pathology, University Hospital Aachen RWTH, Aachen University, Aachen, Germany. [8]Clinical and Transplant Pathology Centre, Institute for Clinical and Experimental Medicine, Prague, Czech Republic. [9]Department of Pathology and Molecular Medicine, Third Faculty of Medicine, Charles University and Thomayer Hospital, Prague, Czech Republic. [10]Department of Hepatogastroenterology, Institute for Clinical and Experimental Medicine, Prague, Czech Republic. [11]Danish Institute of Advanced Study (DIAS), University of Southern Denmark, Odense, Denmark. [12]NNF Center for Protein Research, Faculty of Health Sciences, University of Copenhagen, Copenhagen, Denmark. [13]These authors contributed equally: Sophia C. Mädler, Katrine Holtz Thorhauge, Sophia Steigerwald. ✉e-mail: rosenberger@biochem.mpg.de; mmann@biochem.mpg.de

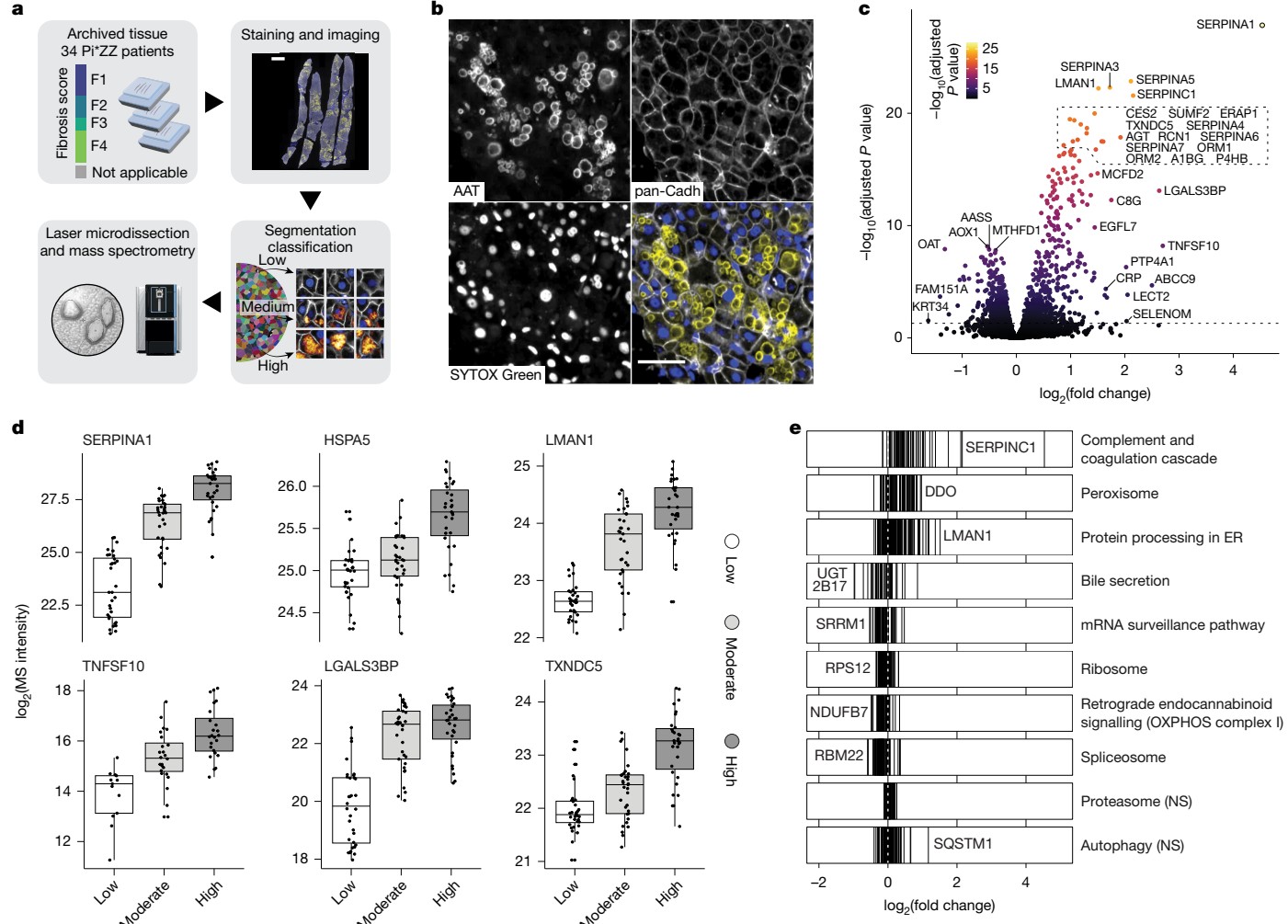

**Fig. 1 | Proteomic mapping of hepatocyte stress response. a**, Overview of the Deep Visual Proteomics workflow. Fibrosis stages are Kleiner scores. **b**, Immunofluorescence staining of AAT, the cell outline marker pan-cadherin (pan-Cadh), nucleus (SYTOX Green) and three-colour overlay. **c**, Proteomic changes in high versus moderate versus low AAT-accumulating cells. Enriched in high on the right side. Top significant and top changed hits are named (paired two-sided moderated *t*-test with load class as covariable, multiple testing corrected; *n* = 96 at 100 shapes per sample). **d**, MS intensity of selected proteins across three classes. One dot is one sample from a patient (*n* = 34). Boxplots show first and third quartiles (box), median (thick line) and whiskers (±1.5 interquartile range). **e**, Significantly (FDR < 0.05) enriched Kyoto Encyclopedia of Genes and Genomes (KEGG) pathways after GSEA. Each line is a member of the pathway. NS, not significant. Scale bars, 1 mm (**a**), 50 μm (**b**).

gene, resulting in misfolding and accumulation of AAT in hepatocytes. Most severe AATD cases are caused by a homozygous Z-variant (Pi*ZZ genotype) with a peak incidence of 1:2,000 in individuals of European descent[1,2,13,14]. Current hypotheses suggest that the severity of liver damage correlates with the amount of accumulated AAT[15–20]. However, the mechanisms driving fibrogenesis or hepatocyte survival versus death remain unclear, leaving potentially druggable targets unexplored.

To address this challenge, we curated a cohort of formalin-fixed paraffin-embedded (FFPE) biopsies and liver explants from patients homozygous for the pathogenic Z-variant, encompassing all fibrosis stages (*n* = 34; Extended Data Fig. 1a and Supplementary Table 1). Despite the same underlying disease-causing mutation at a similar median age (58 ± 10 (s.d.) years) and BMI (25.2 ± 4.0), fibrosis stages varied drastically, indicating unexplored molecular resilience or risk profiles.

## Proteomic map of proteotoxic response

To elucidate the molecular basis of the observed clinical heterogeneity in patients with AATD, we implemented a comprehensive proteomic mapping approach to characterize hepatocyte responses to proteotoxic stress. We first laser microdissected 3-μm-thick FFPE sections from patient biopsies and analysed them with MS following our DVP workflow. After staining for cell outlines and AAT, we segmented and stratified cells into low, moderate and high aggregate load groups on the basis of their microscopy images (Fig. 1a,b). The proteome of 100 shapes—equivalent to the volume of 10–15 complete hepatocytes—was then acquired on the recently introduced Orbitrap Astral mass spectrometer, yielding a high-quality dataset with a mean proteomic depth exceeding 5,000 proteins per sample (Extended Data Fig. 1b,c and Supplementary Table 1). We observed a striking 23-fold difference in AAT levels between low- and high-load cells. The AAT load was captured on the second principal component, preceded only by the fibrosis stage on the first and second component (Extended Data Fig. 1d–f). Given the sparsity of AAT+ cells in biopsy material, this validated our laser microdissection approach as it allowed the biological phenotype to emerge more clearly. Biopsies with a low fibrosis stage exhibited lower AAT baseline loading compared with high fibrosis stages on both proteomics and imaging data, in line with previous findings[15], whereas the maximum load remained fairly equal across

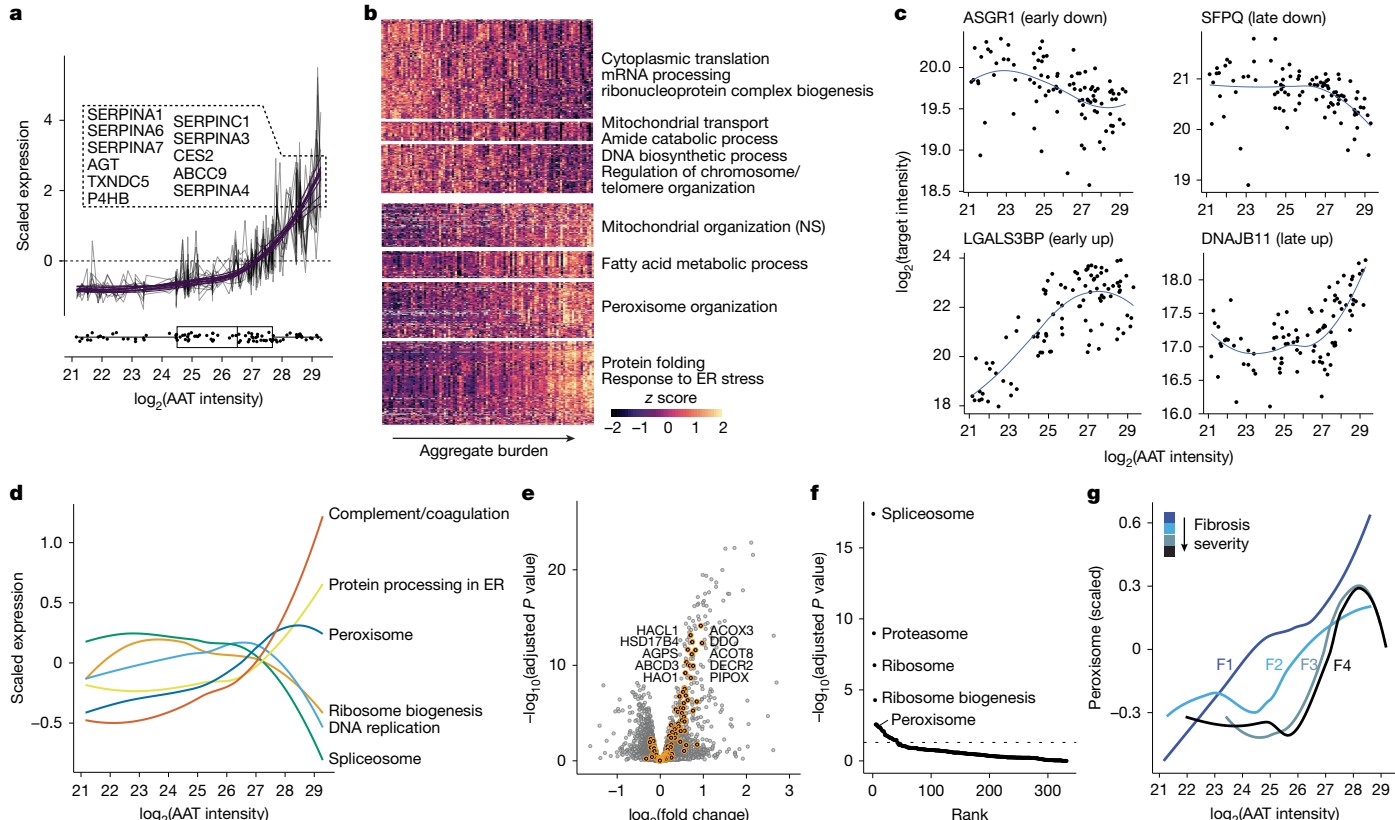

**Fig. 2 | Early and late responses to proteotoxic stress. a**, Expression profile of the top ten proteins correlating with AAT. All DVP samples are plotted, and values belonging to the same protein are on one line. Purple, polynomial fit (third order). Boxplot, distribution of AAT expression values along the $x$ axis. **b**, Clustering of significantly (FDR < 0.01) changed proteins into early and late-responding genes to proteotoxic stress, ordered on $x$ axis by AAT levels. The $y$ axis was broken into seven groups to achieve good coverage of all response types. Significant KEGG terms per box are shown. **c**, Pseudotime expression of top early and late responders by directionality. **d**, Cumulative changes of indicated KEGG pathways expressed as $z$ scores. **e**, Changes in protein levels across three AAT bins, highlighting peroxisomal proteins. Top ten significant hits are named (paired two-sided moderated $t$-test with load class as covariable, multiple testing corrected; $n = 96$). **f**, Top differential functional categories between F1 and F4 fibrotic samples during early AAT accumulation (log$_2$(AAT intensity) <25; two-sided Wilcoxon test, multiple testing corrected). **g**, Cumulative expression of peroxisomal proteins across four fibrosis stages.

all stages (Extended Data Fig. 1g,h). The proteomes of the three load classes differed markedly (17.4% significant hits at <5% false discovery rate (FDR), paired two-sided moderated $t$-test; Fig. 1c). Alongside AAT, several known markers of AATD liver pathology were highly enriched in aggregate-positive cells, such as a 1.6-fold increased endoplasmic reticulum (ER) chaperone HSPA5 and a 2.9-fold increased ER–Golgi cargo receptor LMAN1 (Fig. 1d)[21–23].

Among the most dysregulated hits, we identified other secretory proteins, including many unambiguous SERPINs, coagulation and complement factors (Fig. 1c and Extended Data Fig. 2a–d). This aligns with recent findings of SERPIN sequestration in AAT-inclusions, and supports the notion of crowding in the ER space[18,24], with potential systemic pathological implications due to accumulation of annotated plasma proteins in affected hepatocytes (Extended Data Fig. 2e). Galectin-3 binding protein LGALS3BP and the apoptotic inducer TNFSF10 had the most pronounced positive changes (Fig. 1c,d). LGALS3BP is a hepatocyte-produced protein targeted for secretion that is elevated in plasma from patients with liver disease[25]. Reports describing the immunomodulatory activity of LGALS3BP could explain the involvement of immune cells in AATD liver pathology[15,26,27].

Pathway enrichment analysis showed a strong elevation of proteins related to the three branches of unfolded protein response (UPR) mediated through ATF6, PERK and IRE1 along with a general upregulation of chaperones, accompanied by a reduction in the transcription and translation machinery. This occurred at the expense of physiological functions such as bile secretion (Fig. 1e). Many responses converged into a protective response to reactive oxygen species with upregulation of thioredoxins and glutaredoxins, including an atypical increase in the peroxisomal compartment and reduction of mitochondrial complex I (Fig. 1d and Extended Data Fig. 2a,b,f–j). Proteasomal and autophagy proteins remained largely unchanged, and neither did we detect disturbances of calcium homeostasis (Fig. 1e and Extended Data Fig. 2k).

## Early and late-stage stress responses

Our experimental design, encompassing three aggregate load classes, should allow us to resolve the stepwise progression of molecular events. To determine the sequence in which molecular responses occur during AAT build-up, we first correlated AAT with other protein levels to identify 'followers' that tightly track AAT levels. Proteins of the ER were among the top ten hits, with many destined for secretion (Fig. 2a and Extended Data Fig. 3a). This included many structurally similar SERPINs, and the tight tracking of AAT levels suggests that these proteins accumulate in tandem with AAT rather than being coregulated.

We then categorized proteins into early and late responders to proteotoxic stress caused by AAT accumulation (Fig. 2b and Supplementary Table 2). We observed the most consistent relation with AAT load among coelevated proteins, with most (77.7%) manifesting as late responders and only a smaller fraction as early responders. The immunomodulatory marker LGALS3BP was most prominent among early responders, followed by the ER cargo receptor MCFD2 together with its co-binder LMAN1 (Fig. 2c). A strong peroxisomal biogenesis response

emerged early on, characterized by the peroxisomal proliferation factor PEX11B and other membrane-integral proteins, along with lipid metabolism and superoxide detoxifying proteins (Fig. 2d,e, Extended Data Figs. 3b–d and 4 and Supplementary Table 2). By contrast, most proteins of the core machinery of the UPR appeared later during AAT build-up, despite visual protein accumulation at earlier stages (Fig. 2d and Extended Data Fig. 3e,f). The crosstalk between UPR and peroxisomal activity remains poorly understood, yet lipid metabolism, cholesterol metabolism and reactive oxygen species detoxification intersect both pathways. Together, the data indicate a dominant increase of the ER oxidoreductase-1α (ERO1A)—a main peroxide producer (Fig. 1c and Extended Data Fig. 2f).

We then analysed samples at various fibrosis stages, revealing principal dysregulations with increasing fibrosis stage in proteotoxicity-responsive pathways (Fig. 2f and Extended Data Fig. 5). Notably, this included the peroxisomal response, which showed a gradually prolonged onset time relative to AAT load (Fig. 2g). Peroxisomal chaperones or chaperone-like proteins remained unaltered, suggesting that peroxisomes are unlikely to contribute to the clearance of unfolded proteins (Extended Data Fig. 3d).

## Single-cell mapping in intact tissue

The accumulation of AAT in intact tissue exhibits a pronounced spatial component. Previous work has demonstrated that AAT accumulates unequally along the zonation gradient from portal to central vein axis in patients with AATD with the Pi*ZZ genotype[15,28,29]. Yet, sharp borders and the absence of gradual changes between neighbouring AAT[+] and AAT[−] cells, as well as single positive cells, indicate a more complex picture (Fig. 3a). To map the spatial proteome in these regions, we built on our previous single-cell DVP workflow[5]. We isolated single shapes from selected regions in 10-μm-thick FFPE sections (equivalent to one-third to one-half of a complete hepatocyte) from six F1-stage biopsies. We selected early-stage (F1) biopsies to examine stress processes in a minimally fibrotic environment, reducing potential confounding effects from advanced disease. We quantified the proteome of these 'shapes' one at a time using the Orbitrap Astral mass spectrometer and a variable window precursor selection design (Extended Data Fig. 6a,b).

In this way, we quantified the proteome of 259 single shapes in three biopsies at a median depth of 2,785 proteins, and reaching up to 4,299 proteins (Fig. 3b, Extended Data Fig. 6c,d and Supplementary Table 3). The laser capturing proved highly precise, as evidenced by the complete separation of adjacent AAT[+] and AAT[−] cells (Fig. 3a and Extended Data Fig. 6e–g). On comparing AAT[+] and AAT[−] cells at border regions, we identified similar proteotoxic stress markers as before (Extended Data Figs. 6h and 7a,b). Interestingly, cells of the first or second row within a border region and within their respective AAT class displayed very similar proteomes (Fig. 3c). Consistent with this, the AAT accumulation markers LGALS3BP and ERO1A were markedly different between AAT[+] and AAT[−] cells, but not among first- and second-order neighbours. Consequently, the data support an absence of dedicated stress propagation between neighbouring cells, suggesting that AAT-induced proteotoxic stress is a cell-intrinsic response.

AAT accumulation has been characterized previously as a periportal event[30]. However, our data indicate only partial or no dependence of AAT accumulation on zonation, as evidenced by no or little change in the expression levels of the portal markers ASS1, HAL and ARG1, or the central markers ADH1 and CYP2E1 at borders. We also did not observe any zonation effect in single AAT[+] cells compared with AAT[−] direct neighbours (Extended Data Fig. 7c).

On mapping early- and late-responder markers back onto tissue, we found the expected pattern at border regions for SERPINC1 and LGALS3BP, which mirrored AAT levels early on. The late marker DNAJB11 remained unchanged in four of the six samples, indicating that we captured the accumulation event at an early to medium stage (Fig. 3d).

However, we detected upregulation of the apoptotic inducer TNFSF10 in the border cells in two samples. Further inspection revealed that the aggregate morphology was markedly different, with a globular phenotype in contrast to amorphous AAT accumulation in the other two samples.

## Globular aggregates mark apoptotic cells

Motivated by this observation, we enhanced our DVP workflow to connect morphological information with proteomic data acquisition. We obtained liver resection samples containing thousands of cells with various AAT aggregate morphologies on one slide. After staining and confocal imaging of 3-μm-thick sections of three biological and four technical samples, we segmented cells and transformed the AAT channel signal within cell boundaries into 2,048 features representing AAT morphology using the ConvNeXt convolutional neural network[31]. We projected these representations into a two-dimensional space using uniform manifold approximation and projection (UMAP) and determined 50 equally distributed centre points across the image information layer, from which we selected the 50 closest cells. These were isolated by laser microdissection and measured by MS, resulting in 250 morphology classes representing a total of 12,500 cells (Fig. 4a).

Using UMAP to project the representation of these microdissected cells into a two-dimensional space validated that the convolutional neural network used could indeed stratify cells by aggregate morphologies, with aggregate-devoid cells clustering on one end and globular and amorphous morphologies located at the opposite side and clearly separated from one another (Fig. 4b). We achieved a median proteomic depth of 5,970 proteins from the equivalent of five to ten complete hepatocytes (Extended Data Fig. 8a and Supplementary Table 4). The main drivers of our proteomic data were dynamic changes in keratins and AAT levels on principal components 1 and 2, respectively (Fig. 4c and Extended Data Fig. 8b–d). When grouping samples by proteome into clusters, patient samples were distributed equally across proteomic clusters without apparent genotypic or technical biases (Fig. 4d). As an inverse proof-of-principle, we mapped the proteomic clusters back onto the UMAP image space with clear dimensional separation (Extended Data Fig. 8e). Consistently, samples of one proteome cluster also exhibited the shortest distances to one another on a proteomic UMAP and t-distributed stochastic neighbour embedding plot (Extended Data Fig. 8f,g).

To better understand the molecular responses underlying morphology types, we comparatively analysed samples with clear globular versus amorphous aggregates (Fig. 4e). Contrary to expectation, markers that typically follow AAT levels, such as CES2 and ERO1A, were decreased in globular types. Conversely, the apoptotic inducer TNFSF10 and the inflammatory marker C-reactive protein (CRP) were positively enriched, indicating this to be a late-stage phenotype. We then mapped levels of marker proteins back onto the UMAP-derived image space. Intriguingly, ERO1A and TNFSF10 were localized in two distinct cell populations (Fig. 4f and Extended Data Fig. 9a–d). While ERO1A, indicative of an ongoing UPR response, was highly enriched in amorphous aggregate types, TNFSF10 was present mostly in cells with globular aggregates alongside innate immune system activators. In line with this, gene set enrichment analysis (GSEA) further identified processes related to cell death as upregulated in globular types (Extended Data Fig. 9e).

Given a rather linear response rate of CRP across the image UMAP space (Fig. 4f), we then sorted all samples in pseudotime by CRP expression levels. Across all four biological samples, we observed the emergence and disappearance of small corpuscular aggregates despite retained CRP signal. This was followed by a fulminant amorphous aggregation before condensation into globular aggregates as a late-stage feature before cell death and clearance (Fig. 4g).

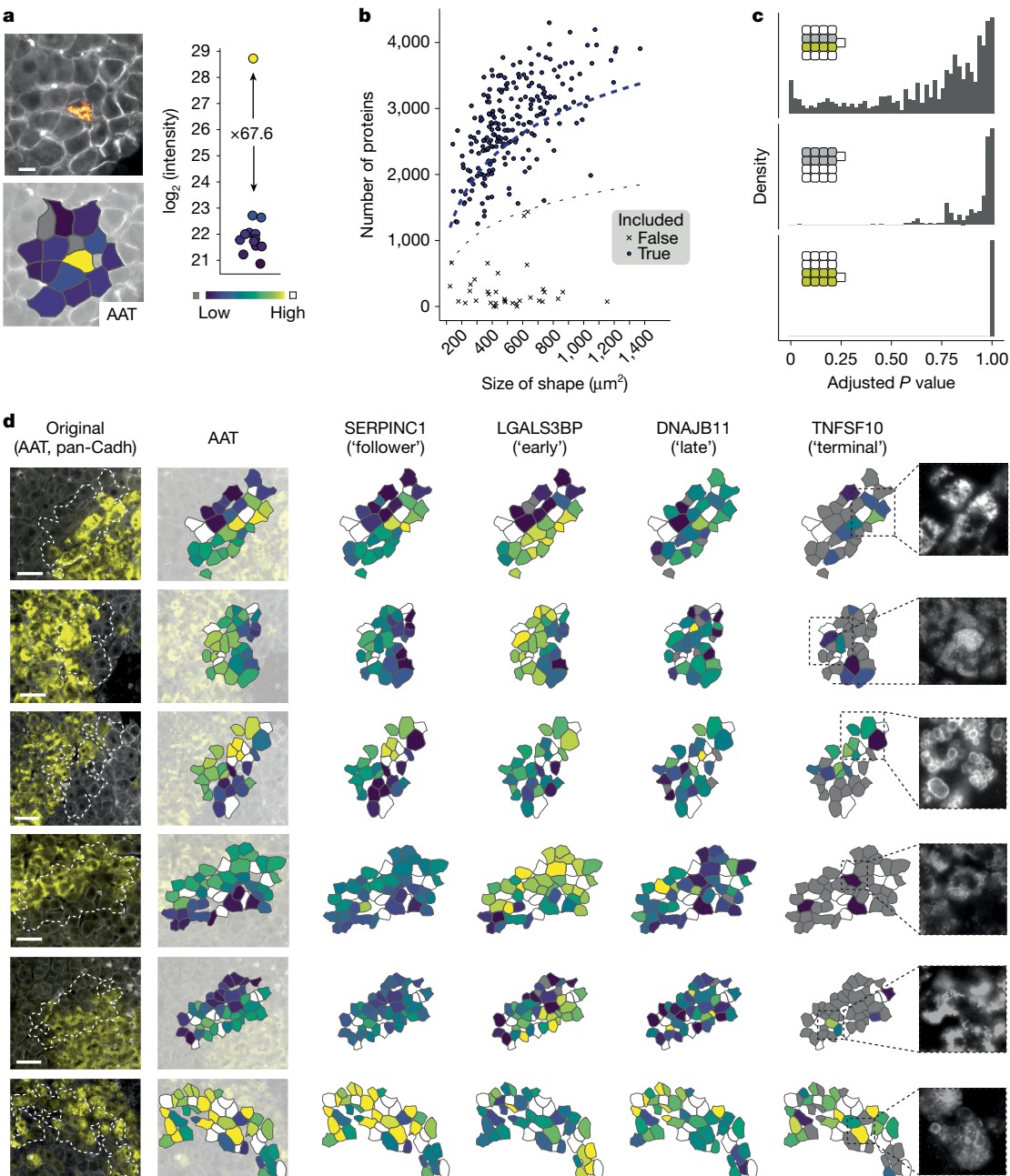

**Fig. 3 | Mapping intact tissue at single-cell level. a,** Enrichment efficiency of the workflow as shown by isolating adjacent cells from FFPE tissue. Proteome quantification of AAT mapped back onto tissue. Boxplot shows AAT expression enrichment. **b,** Number of proteins detected per single shape across all 259 runs against the area of the microdissected shape. Lower grey dotted line marks inclusion cutoff, upper blue dotted line is a logarithmic fit.

**c,** Distribution of $P$ values when comparing single cells at a border (top, $n = 107$), direct AAT⁻ neighbours (middle, $n = 69$) and direct AAT⁺ neighbours (bottom, $n = 111$; two-sided paired moderated $t$-test after multiple testing correction). **d,** Mapping of proteomic information onto the original microscopic image. Cut-out images show AAT staining only. Grey, protein not quantified; white, shape not captured and measured ($N = 6$, $n = 259$). Scale bars, 50 μm.

In addition to TNFSF10, we identified EGF-like domain-containing protein 7 (EGFL7) as a viable marker of this stage that appeared late in the AATD phenotype. Notably, EGFL7 is also upregulated in hepatocellular carcinoma, and high expression levels are associated with poor prognosis[32]. However, a potential link between globular phenotypes and hepatocellular carcinoma incidence in AATD remains unexplored. This late-stage phenotype was further characterized by a stagnating or even declining UPR in late stages, as evidenced by Calreticulin and ERO1A levels, whereas declining levels of proteins such as UGT2B17 suggest the termination of physiological functions in this hepatocyte subtype (Fig. 4h).

## Discussion

We present a pseudotime-resolved proteome of individual hepatocytes undergoing proteotoxic stress due to AAT aggregation. Our findings, derived from FFPE biopsies and resections from patients, provide new insights into the progression and hepatic manifestation in AAT deficiency. Although there are several model systems in the field, including mouse models[33] and patient-derived induced pluripotent stem cells[34], our approach uniquely captures responses to proteotoxic stress directly in patients using human tissue specimens representing the full disease spectrum (stages F1–F4). Notably, our

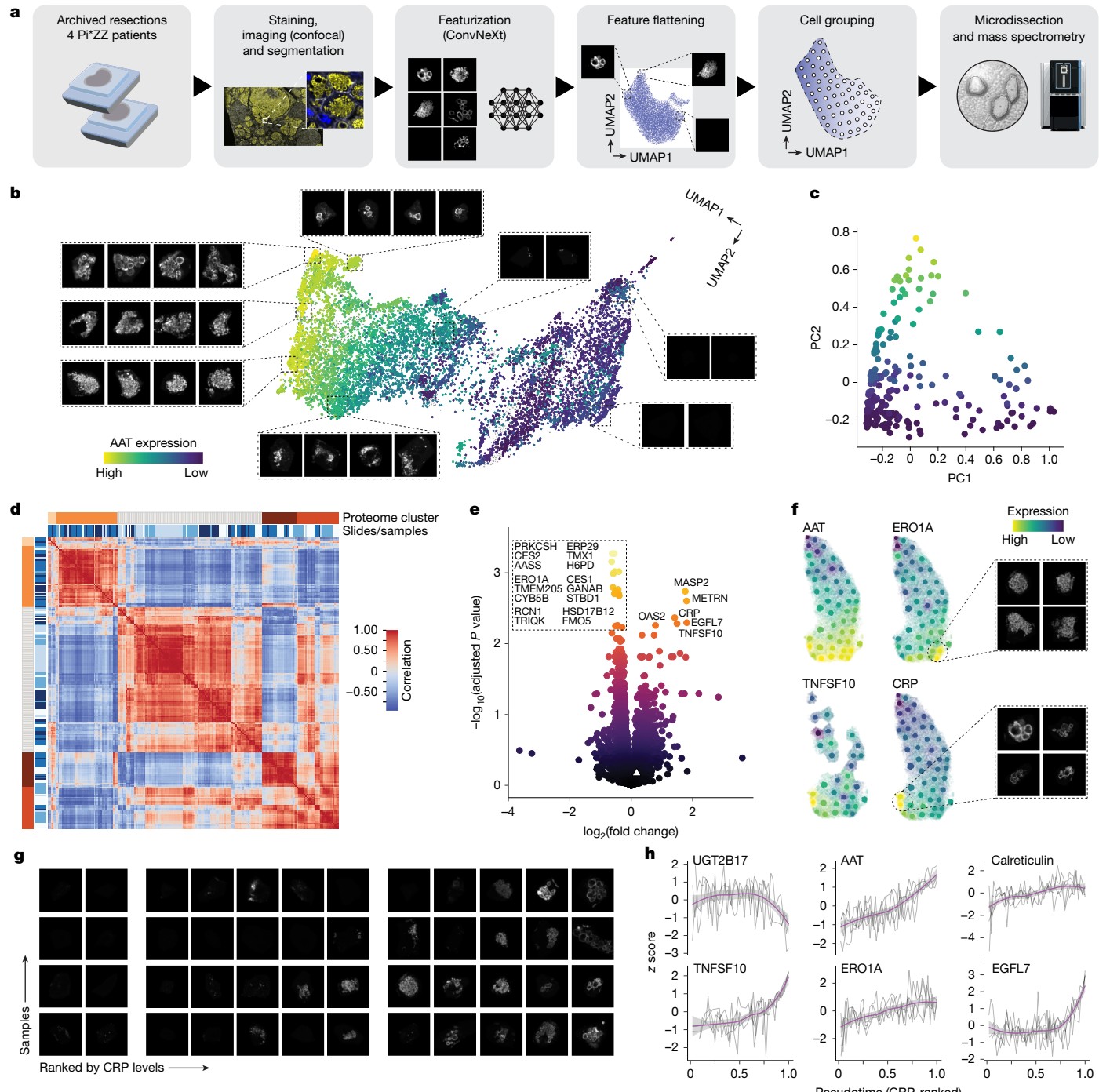

**Fig. 4 | Morphology-guided DVP. a**, Overview of the pipeline. **b**, Projection of all laser microdissected cells (12,500) and representative AAT images in indicated areas. Colour scheme refers to AAT expression level (proteomic). **c**, Proteomic data of 209 samples (after filtering) reduced by PCA (*n* = 4 tissue sections), coloured by AAT expression level. **d**, Proteomic sample correlation heatmap, indicating proteome clusters based on *k*-means clustering (five groups chosen manually) and sample slides. **e**, Comparison of proteomes from cells with globular versus amorphous aggregates after selecting for similar AAT levels (white triangle). Up in globular on the right, top hits annotated (paired two-sided moderated *t*-test after multiple testing correction). **f**, Projection of proteomics data onto image-based UMAP space of one representative sample, with representative images of indicated clusters. **g**, Pseudotime-sorted images of all four biological replicates. Groups mark inflection points of CRP. **h**, Expression levels of indicated proteins in CRP-ranked pseudotime. Each line is one sample, smoothing curve in purple with 95% confidence interval in grey.

data reveal that existing Pi*ZZ models do not accurately recapitulate the UPR, which manifests as a late but fulminant mode of action in our patient-derived samples[1,35]. This discrepancy extends to the globular phenotype, which we now identify as a late-stage cellular feature preceding cell death[16]. Our approach strikingly underlines the power of harnessing patient cohorts and tissues. As many potentially druggable targets and pathways are intrinsically more difficult to validate when appropriate model systems are not in place, this inverts the traditional biomedical discovery cycle. A limitation of this study is the low sample numbers due to limited availability of particularly low-grade fibrotic tissue. This prevents us from further disentangling confounding factors such as alcohol consumption. Nevertheless, the

cellular enrichment by DVP allows the biological phenotype to emerge more clearly, leading to statistically robust and actionable insights even at low sample numbers.

Here we developed a single-cell proteomics approach to generate high-resolution maps of adjacent hepatocytes in intact tissue, leveraging recent advancements in ultra-low-input MS[6,7,36]. Building on our previous work mapping zonation profiles in frozen mouse liver sections at single-cell resolution[5], we now quantify 50% more proteins and apply single-cell DVP (scDVP) to FFPE tissue using the Orbitrap Astral mass spectrometer with a variable window precursor selection scheme. This compatibility with FFPE tissue specimens—the gold standard in diagnostic pathology—expands access to cohorts of virtually any origin, age and size[37], broadening the potential applications of this technology. Spatial transcriptomics has become a powerful tool for spatial analyses in intact FFPE tissue, often approaching single-cell resolution[38]. By contrast, the scDVP approach provides orthogonal biological insights by directly measuring protein abundance with single-cell localization. This is particularly valuable when post-transcriptional regulation and protein accumulation are central to pathology, such as for understanding proteotoxic diseases[38]. Although the scDVP approach is currently limited in throughput compared with transcriptomics, its combination with the herein presented morphology-guided DVP allows efficient sampling of histologically heterogenous material. This could be expanded into morphology-based proteome prediction for large numbers of cells.

Our findings indicate that cells without aggregates are not directly affected or triggered by seeding-like mechanisms from adjacent aggregate-bearing cells. However, the presence of large patches of positive cells implies a propagation mechanism. Given the extensive metabolic perturbations observed, including alterations in fatty acid metabolism and detoxification pathways, AAT aggregate formation in one cell may lead to changes in the metabolic microenvironment, thereby inducing stress and proteostatic imbalance in adjacent cells. This hypothesis aligns with other reports in the AATD field, and similar mechanisms have been proposed in the context of neurodegenerative proteotoxic disorders, where it remains the subject of ongoing debate[39,40].

We present an integration of image featurization and DVP that enables characterization of the entire proteomic and phenotypic lifecycle of stressed hepatocytes in a proteotoxic and fibrogenic liver disease. This methodology establishes a robust framework for dissecting complex cellular processes in situ across a spectrum of proteotoxic diseases. This strategy—an example of digital pathology with quantitative and very deep proteomic readout—yielded exceptionally deep proteomes of 6,000 quantified proteins, sufficient to infer most of the functional proteome of a given cell type. Our datasets are large enough to generate robust models capable of predicting the proteome of a cell solely on the basis of its phenotype. This advancement paves the way for whole-slide proteomics in the future, representing a leap forward in our ability to comprehensively analyse tissue types by MS at exceptional molecular and spatial resolution.

The methods developed here recapitulate known disease progression markers while identifying hundreds of additional dysregulated proteins. The present study is necessarily limited in functional follow-ups, yet these new candidates clearly offer a valuable resource for biological and clinical validation. Of particular clinical relevance, we uncover an early upregulation of the peroxisomal compartment in samples from patients with low-grade liver fibrosis. This response is significantly delayed in high-grade fibrotic samples, suggesting a potential window for therapeutic intervention. Of note, a peroxisomal response is not significantly correlated with fibrotic stages in bulk liver proteomes of patients with alcohol-related liver disease, suggesting that it is specifically important to the AATD pathomechanism[25]. PPAR-α agonists, such as fibrates, which increase peroxisome load in the liver, may be promising candidates for treating patients with late-diagnosed advanced liver fibrosis due to AATD. Given their well-established safety profiles, we suggest that these drugs could be repurposed for AATD, potentially transforming the treatment landscape of this proteotoxic disorder.

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

## Methods

### Clinical cohorts and sample preparation

Patient biopsies and explant samples were obtained at two different sites, Odense University Hospital (OUH) and Aachen RWTH University Hospital (UKA). The sample origin is indicated in Supplementary Table 1. Following ethical guidelines, the clinical data provided here are deidentified by reporting only sample type, fibrosis score and site of origin.

**OUH patient recruitment.** Patients were recruited through the Danish patient organization (Alfa-1 Denmark) and clinical departments for liver and lung diseases as part of a cohort study. The cohort was designed to investigate liver health among nonpregnant adults (minimum age 18 years) diagnosed with AATD of any genotype and carrier status. This specific study includes 16 people diagnosed with Pi*ZZ who consented to undergo the procedure. The study was approved by the Danish Ethical Committee (S-20160187), and participants gave informed consent before enrolment. Participants without a history of liver transplant or decompensated cirrhosis were offered a percutaneous liver biopsy. The patients underwent liver core needle biopsies at OUH between 2017 and 2021. Liver core needle biopsies were taken during this period, stored in 4% formalin and embedded in paraffin. For the assessment of fibrosis stage, FFPE blocks were cut on a microtome into 3-µm-thick sections and mounted on FLEX IHC slides (Dako). Tissue sections were deparaffinized with xylene, rehydrated in serial dilutions of ethanol and stained with Sirius Red. A certified hepatopathologist (S.D.) assessed the Kleiner fibrosis stage (0–4) according to the Pathology Committee of the NASH Clinical Research Network (NAS-CRN).

**UKA patient recruitment.** The recruitment of patients is described in detail in ref. 41. Of this cohort, the present study includes 19 people diagnosed with Pi*ZZ, of whom 14 underwent liver core needle biopsies owing to medical indication and five received a liver transplant because of end-stage liver disease. One patient's sample was later removed owing to its outlier position on the proteome PCA (Supplementary Table 1). Samples were stored in 4% formalin and embedded in paraffin. Fibrosis stage was assessed after trichrome staining of 5-µm-thick sections by a certified hepatopathologist. Blocks were stored at room temperature. Ethical approval was provided by the institutional review board of Aachen University (EK 173/15). All participants provided written informed consent and were treated following the ethical guidelines of the Helsinki Declaration (Hong Kong Amendment) as well as Good Clinical Practice (European guidelines).

### Staining

Polyethylene naphthalate membrane slides (2 µm; MicroDissect GmbH) were exposed to ultraviolet light (254 nm) for 1 h and then coated with Vectabond (Vector Laboratories; catalogue no. SP-1800-7) according to the manufacturer's protocol. FFPE sections (3-µm-thick, DVP, ML; 10-µm-thick, scDVP) were mounted onto these slides and dried at 37 °C overnight. Slides were stored at 4 °C until further processing, upon which slides were baked at 55 °C for 40 min and then deparaffinized and rehydrated (xylene 2 × 2 min, 100% ethanol 2 × 1 min, 90% ethanol 2 × 1 min, 75% ethanol 2 × 1 min, 30% ethanol 2 × 1 min, ddH$_2$O 2 × 1 min). Slides were transferred to prewarmed glycerol-supplemented antigen retrieval buffer (DAKO pH 9 S2367 + 10% glycerol) at 88 °C for 20 min, followed by a 20-min cooldown at room temperature (22 °C). After washing in water, sections were blocked with 5% bovine serum albumin (BSA) in PBS for 1 h, followed by an overnight incubation with primary antibodies in 1% BSA/PBS at 4 °C in a humid staining chamber (1:200 mouse IgG1 monoclonal AAT 2C1, Hycult catalogue no. HM2289; 1:200 rabbit recombinant anti-pan-Cadh (EPR1792Y), Abcam catalogue no. ab51034). After three washes in PBS for 2 min each, secondary antibodies (1:400 goat anti-mouse IgG1, Invitrogen catalogue no. A21127;

1:400 goat anti-rabbit AF647, Invitrogen catalogue no. A21245) in 1% BSA/PBS were applied for 90 min, followed by two 2-min washes in PBS, 15 min in SYTOX Green (1:40,000 in PBS, Invitrogen catalogue no. S7020), and three final 2-min washes in PBS. Excess liquid was removed and samples were coverslipped using SlowFade Diamond Antifade Mountant (Invitrogen, catalogue no. S36963).

### Imaging

**Widefield imaging.** For DVP and scDVP experiments (Figs. 1–3), sections were imaged using a Zeiss Axioscan 7. For all excitation wavelengths (493 nm, 577 nm and 653 nm), 50% light source intensity was used. The illumination time was specified on one section and applied to all consecutive samples within one experimental group. Three z-stacks at an interval of 2 µm were recorded with a Plan-Apochromat ×20, 0.8 numerical aperture M27 objective and an Axiocam 712 camera at 14-bit, with a binning of 1 and a tile overlap of 10%, resulting in a scaling of 0.173 µm × 0.173 µm. Multiscene images were then split into single scenes, z-stacks combined into a single plane using extended depth of focus (variance method, standard settings) and stitched on the pan-Cadh channel using the proprietary Zeiss Zen Imaging software.

**Confocal imaging.** For experiments with downstream machine learning applications (Fig. 4), sections were imaged on a Perkin Elmer Opera-Phenix high-content microscope, controlled with Harmony v.4.9 software, at ×40 magnification and 0.75 numerical aperture, with a binning of 1 and a per tile overlap of 10%. Only one z-plane was recorded, which was specified manually for each slide and channel. The three channels were imaged consecutively after deactivation of simultaneous recording to avoid any leakage between channels.

### Cell selection with Biological Image Analysis Software

Images were imported as .czi files into the Biological Image Analysis Software (BIAS) using the packaged import tool[4]. Within BIAS, images were then retiled to 1,024 × 1,024 pixels with an overlap of 10%, and empty tiles were excluded from further analyses. Outlines of all cells per biopsy were identified in an unbiased way by using Cellpose v.2.0 with the default cyto2 model based on anti-pan-Cadh stains[42]. Masks were imported into BIAS, and duplicates, as well as cells touching the borders of a tile (0.1% on each side), were removed. Further filtering was applied to retain cells with a minimum size of 3,000 pixels, enriching for the hepatocyte population. For classification based on low, medium and high aggregate load, all cells per biopsy or explant tissue were divided into five classes using a multilayer perceptron with the following parameters: weight scale 0.01; momentum 0.01; maximum iterations 10,000; epsilon 0.0005 and five neurons in the hidden layer. Classification was based on the AAT maximum, median and mean intensity within the cell outline mask, involving no human intervention. The low class was attributed to the cells with the lowest normalized mean intensity, medium to the third highest and high to the highest normalized mean intensity; the other two intermediate classes were dropped. Reference points were selected on the basis of prominent nuclear and histological features; 100 cells were picked randomly for excision.

For single shape experiments, six characteristic low fibrosis samples (all F1) and regions were selected that presented with a clear border-like phenotype (that is, a row of AAT+ cells in direct neighbourhood to AAT− cells) or with single AAT+ cells surrounded by AAT− cells. The cells were selected manually in BIAS, starting from the innermost cell and moving spiral-like to the outermost cell, thus avoiding cross-contamination of consecutively cut material.

### Single-cell image generation

Images were flat-field corrected during image acquisition using the Perkin Elmer Harmony software (v.4.9). Stitching of the flat-field corrected image tiles was performed using the Python library scPortrait (https://github.com/MannLabs/scPortrait). The stitched tile positions

were calculated using the anti-pan-Cadh stains imaged in the Alexa647 channel as a reference and then transferred to the other image channels. During stitching, the tile overlap was set to 0.1, the filter sigma parameter to 1 and the max shift parameter to 50.

The stitched images were then further processed in scPortrait. Cell outlines were identified on the basis of the seven times downsampled anti-pan-Cadh stains using Cellpose v.2.0 with the pretrained 'cyto' model[42]. Segmentation was performed in a tiled mode with a 100-pixel overlap. After resolving the cell outlines from overlapping regions, the resulting segmentation mask was upscaled to the original input dimensions during which the edges of the masks were smoothened by applying an erosion and dilation operation with a kernel size of 7.

Then, the generated segmentation mask was used to extract single-cell image datasets with a size of 280 pixels × 280 pixels. During extraction, the same single-cell image masks are used to obtain the pixel information from each channel for each cell. The resulting single-cell images were then rescaled to the [0, 1] range while preserving relative signal intensities. The resulting single-cell image datasets were filtered to contain only cells from within manually annotated regions in the tissue section containing hepatocytes but not fibrotic tissue.

### Cell selection with the convolutional neural network

The filtered single-cell image datasets produced by scPortrait were further filtered to remove any cells that fell outside the 5–97.5% size percentile. Representations of the remaining cells were generated by featurization using the natural image-pretrained ConvNext model[31]. For this, the single-cell images depicting the Alpha-1 channel were rescaled to the expected image dimensions of $N$ pixels × $N$ pixels and triplicated to generate a pseudo-rgb image. Inference was then performed using the huggingface transformers package v.4.26 (ref. 43).

The resulting 2,048 image features were projected into a two-dimensional space using the UMAP algorithm[44]. The UMAP dimensions were calculated on the basis of the first 50 principal components and the 15 nearest neighbours. Using the spectral clustering algorithm from scikit-learn[45], the resulting UMAP space was split into 50 clusters. The geometric centre of each cluster was calculated and the 50 cells with the smallest Euclidean distance to the cluster centre were selected for laser microdissection.

Contour outlines of the selected cells were generated in scPortrait using the py-lmd package[46], whereby the cell outlines were dilated with a kernel size of 3 and a smoothing filter of 25 was applied. Furthermore, the number of points defining each shape were compressed by a factor of 30 to improve laser microdissection cutting performance. The cutting path, that is, which cell is cut after one another, was optimized using the Hilbert algorithm (https://github.com/galtay/hilbertcurve).

### Laser microdissection

After aligning the reference points, contour outlines were imported, and shapes were cut using the LMD7 (Leica) laser microdissection system in a semi-automated mode with the following settings: power 45; aperture 1; speed 40; middle pulse count 1; final pulse 0; head current 42–50%; pulse frequency 2,982 and offset 190. The microscope was operated with the LMD beta v.10 software, calibrated for the gravitational stage shift into 384-well plates (Eppendorf, catalogue no. 0030129547), leaving the outermost rows and columns empty. To prevent sorting errors, a 'wind shield' plate was placed on top of the sample stage. Plates were then sealed, centrifuged at 1,000g for 5 min, and subsequently frozen at −20 °C for further processing.

### Peptide preparation and Evotip loading

Peptides were prepared as described previously using a BRAVO pipetting robot (Agilent)[47]. Briefly, 384-well plates were thawed, and shapes (both combined and individual) were rinsed from the walls into the bottom of the well with 28 µl of 100% acetonitrile (ACN). The wells were dried completely in a SpeedVac at 45 °C, followed by the addition of 6 µl of 60 mM triethylammonium bicarbonate (Supelco, catalogue no. 18597) (pH 8.5) supplemented with 0.013% n-dodecyl-beta-D-maltoside (Sigma-Aldrich, catalogue no. D5172). Plates were sealed and incubated at 95 °C for 1 h. After adjusting to 10% ACN, samples were incubated again at 75 °C for 1 h. Subsequently, 6 ng and 4 ng of trypsin and Lys-C protease, respectively, in 1 µl of 60 mM triethylammonium bicarbonate buffer were added to each sample, and proteins were digested for 16 h at 37 °C. The reaction was quenched by adding trifluoroacetic acid to a final concentration of 1%. Peptide samples were then frozen at −20 °C.

For loading, new Evotips were first soaked in 1-propanol for 1 min, then rinsed twice with 50 µl of buffer B (ACN with 0.1% formic acid). After another 1-propanol soaking step for 3 min, the tips were equilibrated with two washes of 50 µl buffer A (0.1% formic acid). Samples were loaded into 70 µl of preloaded buffer A. Following one additional buffer A wash, the peptide-containing C18 disk was overlaid with 150 µl buffer A and centrifuged briefly through the disk. All centrifugation steps were performed at 700g for 1 min. The final tips were stored in buffer A for a maximum of 4 days before liquid chromatography (LC)-MS.

### LC-MS data acquisition

The peptide samples were analysed using an Evosep One LC system (Evosep) coupled to an Orbitrap Astral mass spectrometer (Thermo Fisher Scientific). Peptides were eluted from the Evotips with up to 35% ACN and separated using an Evosep low-flow 'Whisper' gradient for DVP samples, or an experimental Evosep 'Whisper Zoom' gradient for single shapes and DVP-machine learning samples, with a throughput of 40 samples per day on an Aurora Elite TS column of 15-cm length, 75-µm-internal diameter, packed with 1.7 µm C18 beads (IonOpticks). The column temperature was maintained at 50 °C using a column heater (IonOpticks).

The Orbitrap Astral mass spectrometer was equipped with a FAIMS Pro interface and an EASY-Spray source (both Thermo Fisher Scientific). A FAIMS compensation voltage of −40 V and a total carrier gas flow of 3.5 l min$^{-1}$ were used. An electrospray voltage of 1,900 V was applied for ionization, and the radio frequency level was set to 40. Orbitrap MS1 spectra were acquired from 380 to 980 $m/z$ at a resolution of 240,000 (at $m/z$ 200) with a normalized automated gain control (AGC) target of 500% and a maximum injection time of 100 ms.

For the Astral MS/MS scans in data-independent acquisition (DIA) mode, we determined the optimal methods experimentally across the precursor selection range of 380–980 $m/z$: (1) for DVP samples, a window width of 5 Th, a maximum injection time of 10 ms and a normalized AGC target of 800% were used. (2) For DVP-machine learning samples, a window width of 6 Th, a maximum injection time of 13 ms and a normalized AGC target of 500% were applied. (3) For single shapes and other DIA scans, the window width was optimized on the basis of precursor density across the selection range of 380–980 $m/z$. A total of 45 variable-width DIA windows (Supplementary Table 3) were acquired with a maximum injection time of 28 ms and an AGC target of 800%. The isolated ions were fragmented using higher-energy collisional dissociation with 25% normalized collision energy. Detailed method descriptions are provided in a default format with each supplementary data table.

### Spectral searches and normalization

The raw files were searched together with match-between run-in library-free mode within each experimental group with DIA-NN v.1.8.1 (ref. 48). A FASTA file containing only canonical sequences was obtained from Uniprot (20,404 entries, downloaded on 2 January 2023), and the disease-causing amino acid was changed manually (E342K). We allowed a missed cleavage rate of up to 1, and set mass accuracy to 8, MS1 accuracy to 4 and the scan window to 6. Proteins were inferred on the basis of genes, and the neural network classifier was set to 'single-pass mode'. For DVP and DVP-machine learning samples, precursor intensities in

the 'report.tsv' file were then normalized using the directLFQ GUI at standard settings including a minimum number of non-NaN ion intensities required to derive a protein intensity of 1 (ref. 49). The single shape data was additionally median-normalized to a set of proteins quantified across all samples (451 proteins quantified in 100% of included samples; Supplementary Table 3), thereby correcting for the dependence of protein numbers on shape size[5].

## Data analysis and statistics

Data were analysed using R v.4.4.1. The directLFQ output file 'pg_matrix.tsv' was used for all subsequent data analysis, including the reported protein counts. Samples were included if the number of protein groups exceeded (1) the mean − 1.5 s.d. for DVP, resulting in 5.9% (6 of 102) dropouts; (2) the mean − 0.5 s.d. for DVP-machine learning samples; (3) a fitted logarithmic curve − 1.5 interquartile ranges for scDVP, taking the relation between size and proteomic depth into account, resulting in 15.4% (40 of 259) dropouts. The lower cutoffs were selected after manual inspection of the data distribution. Although some samples were collected in technical duplicates per patient biopsy, only the first replicate was used for statistical analyses and all reported measurements were taken from distinct samples. Coefficients of variation were calculated on nontransformed intensity values. For principal component analysis (PCA), the R package PCAtools v.2.16.0 was used on a complete data matrix, removing the lower 10% of variables based on variance. Statistical analyses were performed on proteins with at least 30% data completeness across samples, assuming normality using the limma package v.3.60.3 with two-sided moderated $t$-tests and 'fdr' as a multiple testing correction method. A per patient statistical pairing was applied for DVP and single shape experiments. Intensity and fold changes are reported as $\log_2$-transformed values unless indicated otherwise. GSEA was conducted using WebGestalt 2024 against the indicated databases, with an FDR of <0.05 considered significant[50]. Interaction networks were calculated with STRING database at standard settings[51]. Plasma proteins were retrieved from the Human Protein Atlas resource section with the search term 'sa_location:Secreted to blood AND tissue_category_rna:liver;Tissue enriched'[52]. The timing of responses ranked by the absolute difference between B values of limma's moderated $t$-test comparing three AAT load groups: low to moderate, and moderate to high. Only proteins with more than 70% data completeness and significance (FDR < 0.05) in either or both comparisons were considered. Differential pathway expression across fibrosis stages was calculated by fitting a linear model through $\log_2$-transformed intensity values of individual proteins in samples with $\log_2$(AAT)-intensity <25, and the slopes of proteins in a particular pathway were compared between F1 and F4 samples by a two-sided Wilcoxon rank test without assumption of normality. Indicated $P$ values are corrected for multiple testing using the 'fdr' method. Spatial data was mapped using the 'simple features' package. Binned expression presented in supplementary tables was constructed by grouping AAT or CRP expression into ten equidistant bins and on median expression of proteins across samples in each bin.

## Reporting summary

Further information on research design is available in the Nature Portfolio Reporting Summary linked to this article.

## Data availability

The MS proteomics data have been deposited to the ProteomeXchange Consortium through the PRIDE[53] partner repository with the dataset identifier PXD054440. Imaging data of explant and morphological clusters have been deposited to BioStudies[54] with the identifier S-BIAD1523.

## Code availability

The R and Python code used in this study are documented at https://github.com/MannLabs/Proteotoxicity with a readily deployable script to generate most of the figure panels.

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

**Acknowledgements** We thank our colleagues at the Department of Proteomics and Signal Transduction at the Max Planck Institute of Biochemistry as well as our colleagues at the Centre for Proteome Research in Copenhagen for their input and support. We are particularly grateful for the technical assistance of D. Wischnewski (MPIB), for great scientific discussions with A. Wilson and J. Kaserman (Boston University School of Medicine) and for valuable input from T. Nordmann (MPIB), M. Thielert (MPIB), V. Brennsteiner (MPIB), L. Grauvogel (MPIB), A. Sinha (MPIB), K. Madden (MPIB) and S. Haber (UK Aachen). We thank the Computing Centre and the Imaging Facility of the MPI of Biochemistry for their support and resources. F.A.R. is an EMBO postdoctoral fellow (ALTF 399-2021). S.C.M. is a PhD fellow of the Boehringer Ingelheim Fonds. K.H.T. received a travel grant from the OUH Internationalisation Fund. This study has been supported by the Horizon-2020 under the MICROB-PREDICT programme (M.M., A.K., no. 825694); by the Max Planck Society for Advancement of Science (M.M.); by a grant from the Alpha-1 Foundation (F.A.R.) and Alfa-1 Liver Study (A.K.); by the Deutsche Forschungsgemeinschaft DFG through SFB 1382 (P.S., ID 403224013); P.S. holds a Heisenberg professorship (STR1095/6-1); P.B. is supported by the German Research Foundation (DFG, Project IDs 322900939 and 445703531), European Research Council (ERC Consolidator grant no. 101001791) and the Federal Ministry of Education and Research (BMBF, STOP-FSGS-01GM2202C).

**Author contributions** Conceptualization: F.A.R., K.H.T., S.C.M., P.S. and M.M. Project team leads: S.C.M., K.H.T. and S.S. Methodology: F.A.R., S.C.M. and S.S. Software: S.C.M., M.L. and N.A.S. Validation: F.A.R., C.A.M.W., M.O., M.W. and M.Z. Formal analysis: F.A.R., S.C.M. and M.L. Investigation: F.A.R., S.C.M., K.H.T., S.S., M.L., C.A.M.W., M.W., A.M. and M.Z. Resources: K.H.T., M.F., S.D., P.B., O.F., S.F., A.K., P.S. and M.M. Data curation: F.A.R. and S.C.M. Writing—original draft: F.A.R. and M.M. Writing—review and editing: all authors. Visualization: F.A.R., S.C.M. and M.L. Supervision: F.A.R., P.S. and M.M.

**Funding** Open access funding provided by Max Planck Society.

**Competing interests** M.M. is an indirect investor in Evosep. A patent for treatment of conditions related to α1-antitrypsin deficiency with PPARα agonists has been filed with the European Patent Office (application number EP24205578.8). The other authors declare no competing interests.

**Additional information**
**Correspondence and requests for materials** should be addressed to Florian A. Rosenberger or Matthias Mann.

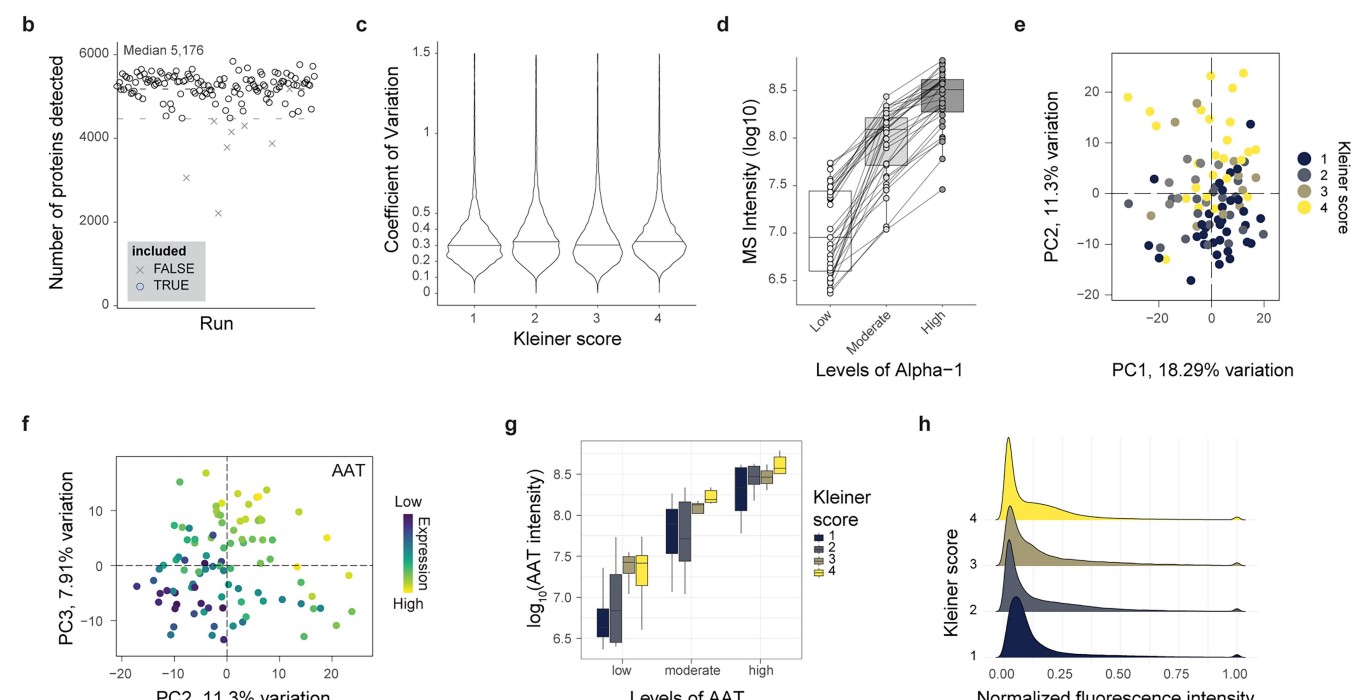

| Kleiner score | F1 | F2 | F3 | F4 | Resections | unknown |
|---|---|---|---|---|---|---|
| Genotype | Pi*ZZ (100%) | | | | | |
| Number | 11 | 7 | 3 | 5 | 5 | 3 |
| Female sex | 45.5% (5/11) | 0% (0/7) | 33.3% (1/3) | 60.0% (3/5) | 100% (3/3, 2 NA) | 33.3% (1/3) |
| Age at biopsy (y) | 58.0 (±11.1) | 54.9 (±11.0) | 64.0 (±10.9) | 56.5 (±4.5) | 56.4 (±7.5) | 63.3 (±7.6) |
| Alcohol | 18.1% (2/11) | 33.3% (2/6, 1 NA) | 0% (0/3) | 50.0% (2/4, 1 NA) | 0% (0/4, 1 NA) | 33.3% (1/3) |
| Diabetes | 3.0% (1/33, 1 NA) | | | | | |
| BMI | 24.7 (±2.92) | 25.6 (±3.1) | 24.4 (±2.99) | 25.5 (±1.88) | 28.0 (±3.6) | 31.3 (±8.7) |

**Extended Data Fig. 1 | Quality control of Deep Visual Proteomics data.**
**a**, Summary of clinical metadata shown as patient numbers or percentages (absolute numbers in brackets). Values reported as mean ± SD. **b**, Number of proteins detected across all runs before excluding technical replicates (n = 134). Upper dotted line: median number of protein groups. Lower dotted line: median − 1.5 SD. Excluded samples are marked with crosses. **c**, Coefficient of variation across fibrosis stages. **d**, MS intensity of alpha-1 antitrypsin in the three microdissected cell classes (N = 34 patients, n = 96 samples). **e**, Principal component analysis showing components 1 and 2 colored by fibrosis stage, and

**f**, components 2 and 3 colored by alpha-1 antitrypsin level. Each dot represents one sample (n = 96). **g**, Alpha-1 antitrypsin levels by fibrosis stage across the three microdissected cell classes (N = 31 patients, n = 88 samples with known fibrosis status). **h**, AAT fluorescence intensity distribution across cells from biopsies with Kleiner scores (n = 31 biopsies, 2,967,275 cells total). Values were normalized to a 0–1 range after removing outliers (below 1st and above 99th percentile). Box plots show first and third quartiles (box), median (thick line), and whiskers (±1.5 interquartile range).

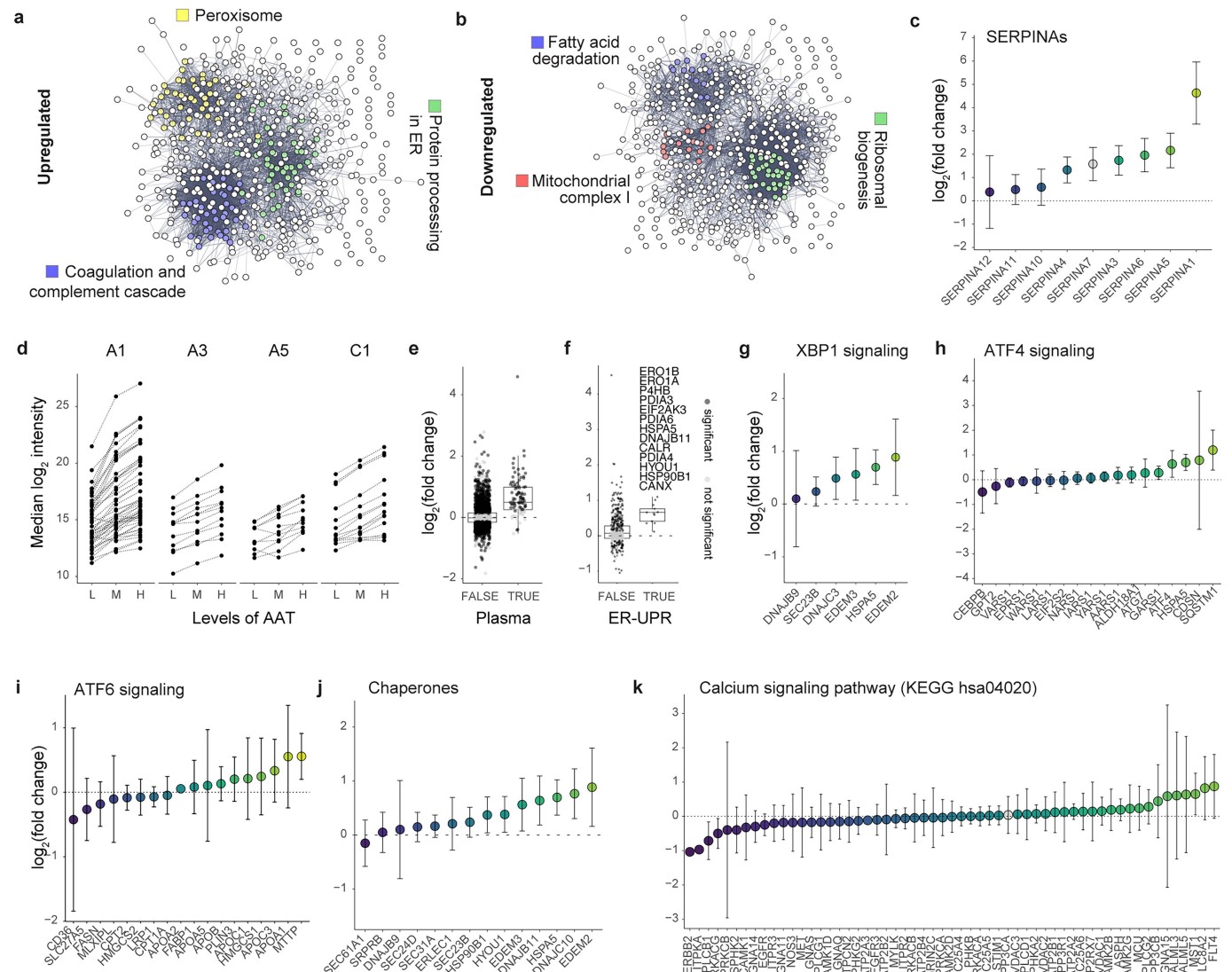

**Extended Data Fig. 2 | Proteomics responses to proteotoxic stress.**
**a**, STRING interaction network of significantly upregulated proteins (FDR < 0.05) and **b**, downregulated proteins in cells (see Fig. 1c). **c**, Changes in SERPINA protein family members relative to baseline hepatocyte group. **d**, Changes in SERPIN protein family precursors across three hepatocyte classes. Lines connect the same precursor (defined as peptide by charge state). **e**, Changes in proteins targeted for plasma secretion relative to baseline hepatocyte group. Plasma protein dataset obtained from Human Protein Atlas using query "sa_location: Secreted to blood AND tissue_category_rna:liver;Tissue enriched" ('FALSE'

includes 5806 protein, 'TRUE' includes 100 proteins). **f**, Changes in ER proteins (annotated as such in Uniprot, n = 677) relative to baseline hepatocyte group with a manually curated subset of ER-UPR components. **g-k**, Protein levels in indicated pathways comparing cells with and without aggregates. Circles show means, bars indicate SD across patient samples (n = 34). Proteins in panels i-m were manually selected; panel n shows proteins from KEGG database. Box plots show first and third quartiles (box), median (thick line), and whiskers (±1.5 interquartile range).

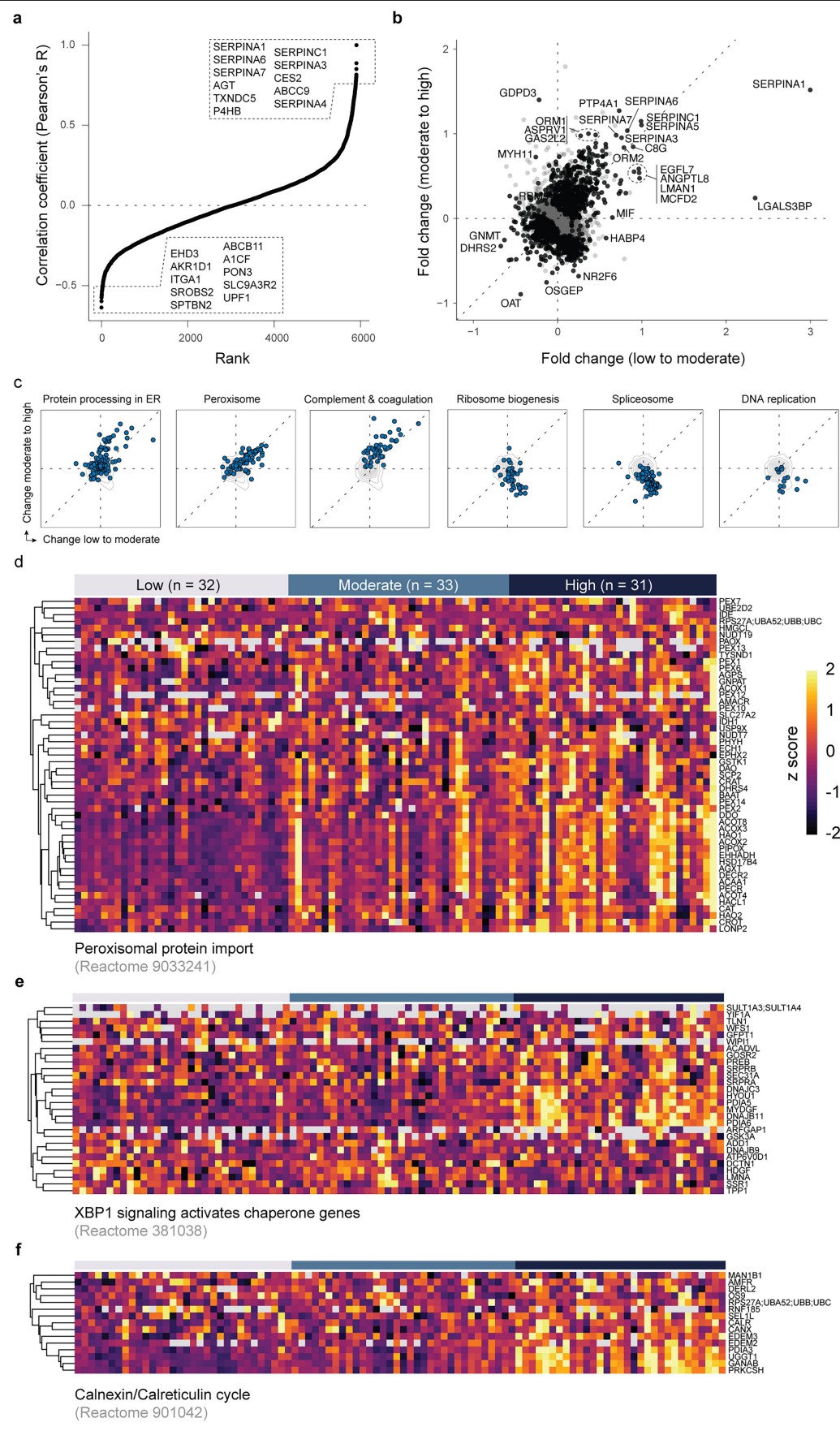

**Extended Data Fig. 3** | See next page for caption.

**Extended Data Fig. 3 | Early and late responses to proteotoxic stress.**
**a**, Pearson's correlation coefficient (R) between each detected protein and alpha-1 antitrypsin levels per MS sample. Top and bottom 10 proteins are highlighted in boxes. **b**, Proteomic changes across high, moderate, and low AAT-accumulating cells with manually curated labels. **c**, Panel (b) overlaid with proteins from indicated KEGG pathways. Non-pathway proteins shown as density cloud. **d**, Expression levels of proteins involved in peroxisomal protein import, **e**, XBP1 signaling, and **f**, the Calnexin/Calreticulin cycle. Values shown as z-scores (assuming normal distribution) across samples split by load class. Database identifiers listed below each graph (n = 96 samples from 34 patients).

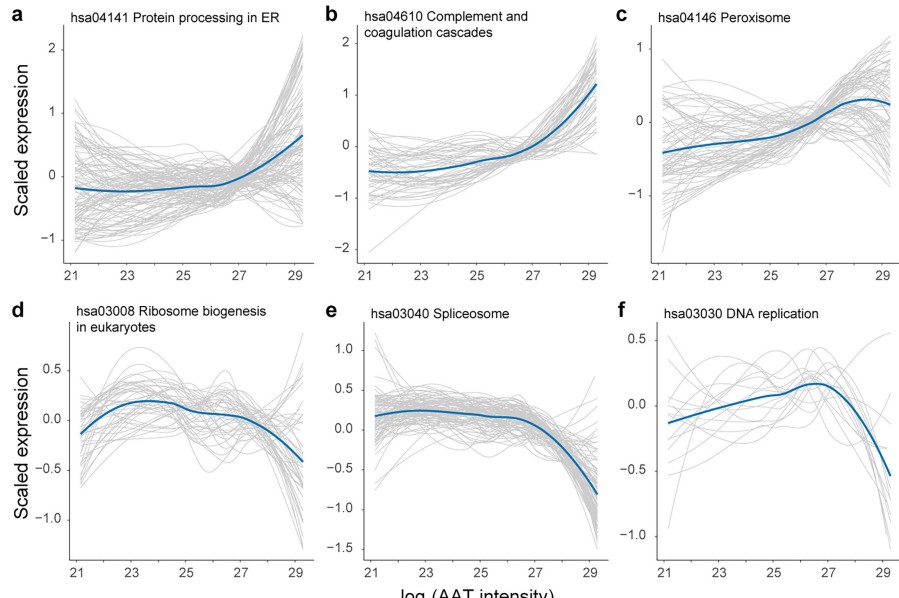

**Extended Data Fig. 4 | Changes in functional pathways. a-f**, Scaled protein intensities (z-scored) from indicated KEGG pathways plotted against AAT intensity. KEGG pathways identified by 'hsa00000' identifiers. Purple lines show local regression (span 0.75, degree 2).

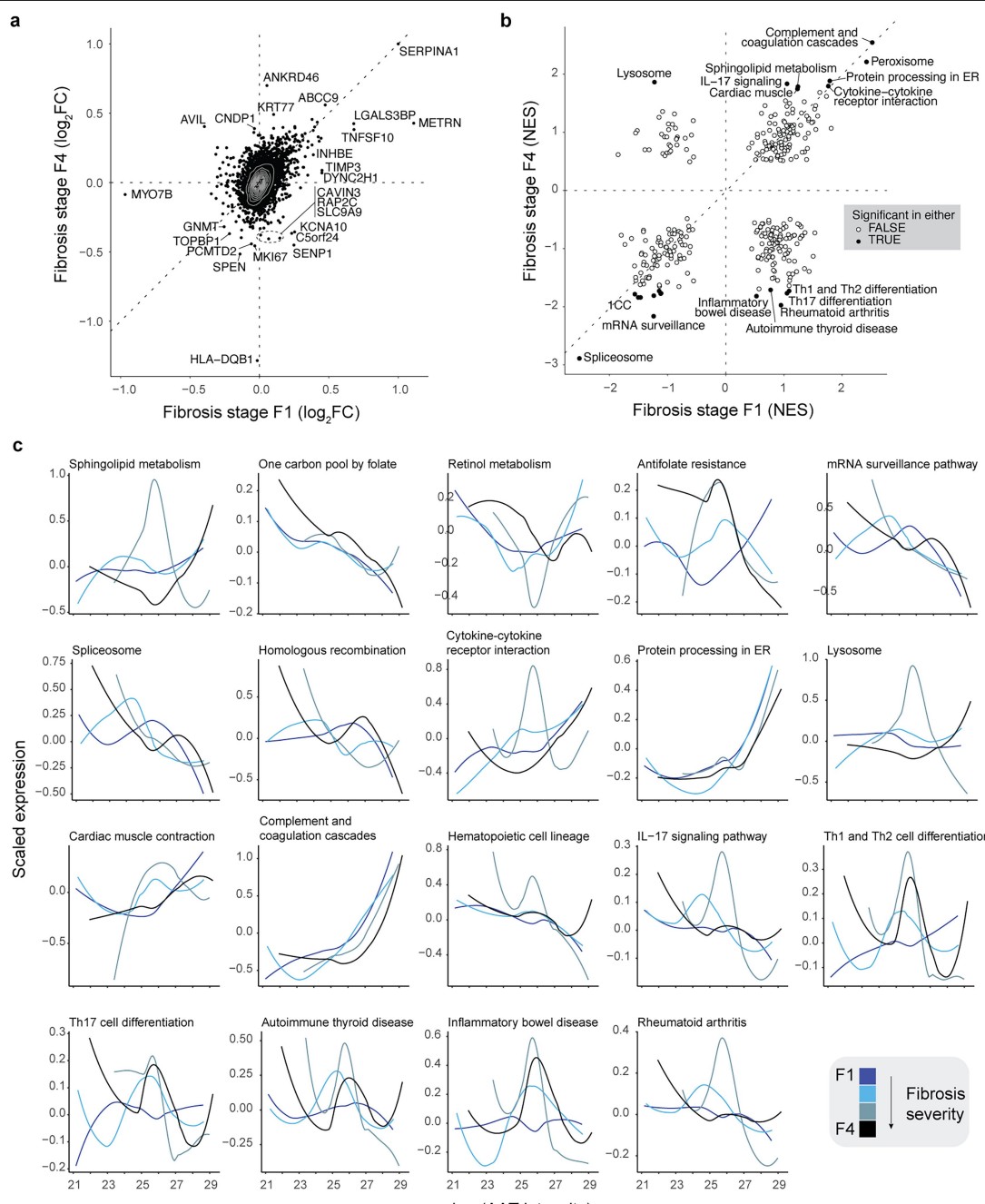

**Extended Data Fig. 5 | Impact of fibrosis on functional pathways in relation to AAT load. a**, Statistical comparison of three AAT load-defined hepatocyte classes, stratified by fibrosis grade (F1: n = 11, F4: n = 6; paired two-sided t-test with load class as covariate, multiple testing corrected). **b**, Gene Set Enrichment Analysis of fold changes shown in panel (a). Significant pathways are indicated by filled (black) or unfilled (white) markers; selected non-redundant terms are labeled. **c**, Scaled protein intensities (z-scored within fibrosis groups) for detected proteins in KEGG pathways plotted against AAT intensity. KEGG identifiers shown as 'hsa00000'. Purple line indicates local regression (span 0.75, degree 2). Legend for all panels shown in top right.

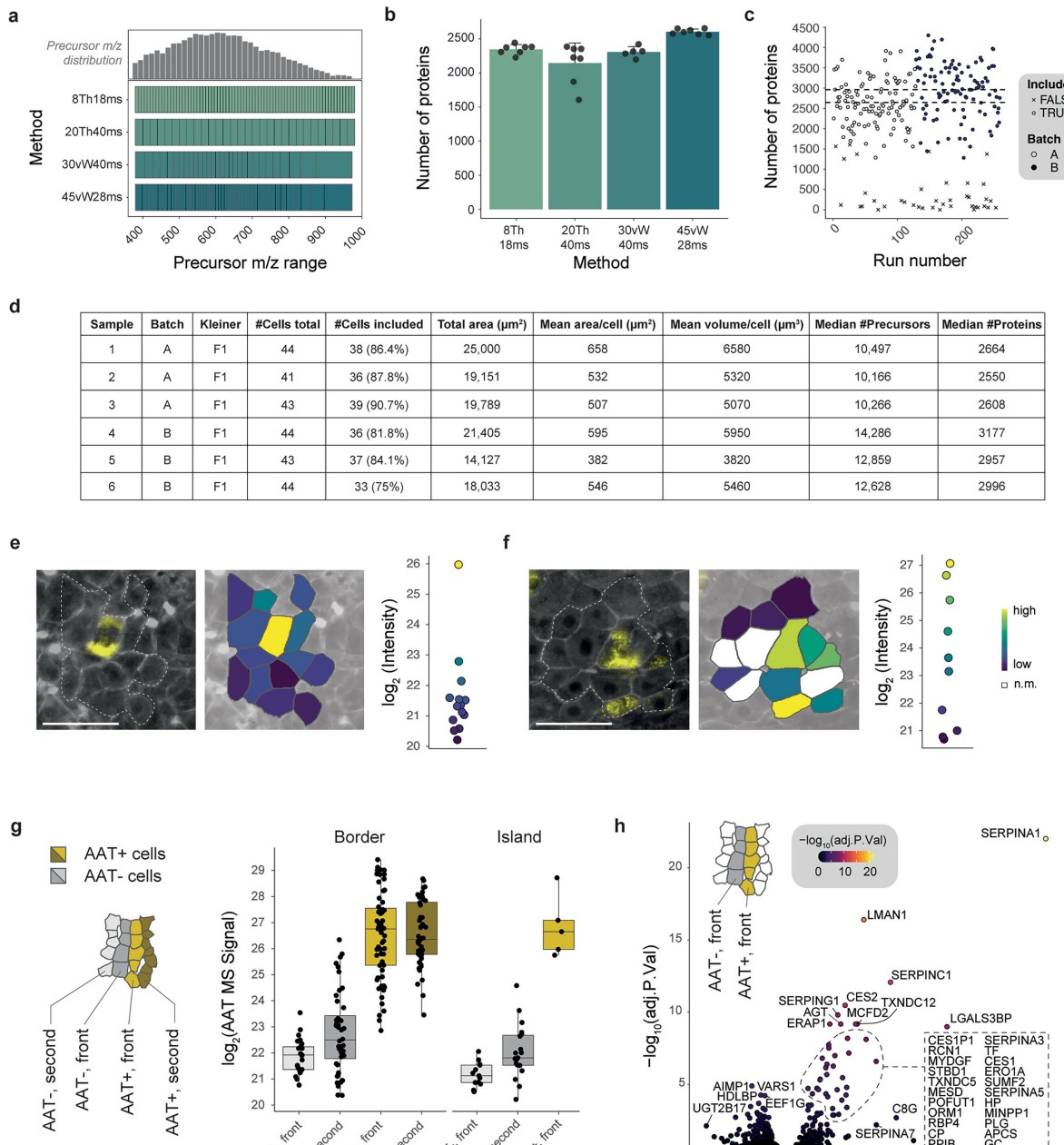

**Extended Data Fig. 6 | The single-cell proteome. a**, MS/MS acquisition design on the Orbitrap Astral mass spectrometer showing window width and injection time. "v" indicates variable windows (represented by box sizes). AAT expression shown by color in regions with single-positive cells. AAT levels for indicated shapes displayed in adjacent dot plots. **b**, Number of proteins quantified per acquisition strategy (n > 5) after exclusion of samples with less than 500 proteins. Error bars are positive standard deviations. **c**, Protein quantification across all hepatocyte shape runs (n = 259). Lower dashed line: median in batch A (2601 proteins); upper dashed line: median in batch B (3004 proteins). **d**, Summary statistics per sample. **e**, and **f**, AAT expression visualized by color in regions with single-positive cells. AAT levels for indicated shapes shown in adjacent dot plots. **g**, AAT expression in specified regions measured by immunofluorescence across all included samples (n = 219). Box plots show first and third quartiles (box), median (thick line), and whiskers (±1.5 interquartile range). **h**, Statistical comparison between AAT+ and AAT- cells in regions classified as 'borders' (paired two-sided t-test, multiple testing corrected, 63 AAT+ cells and 44 AAT- cells).

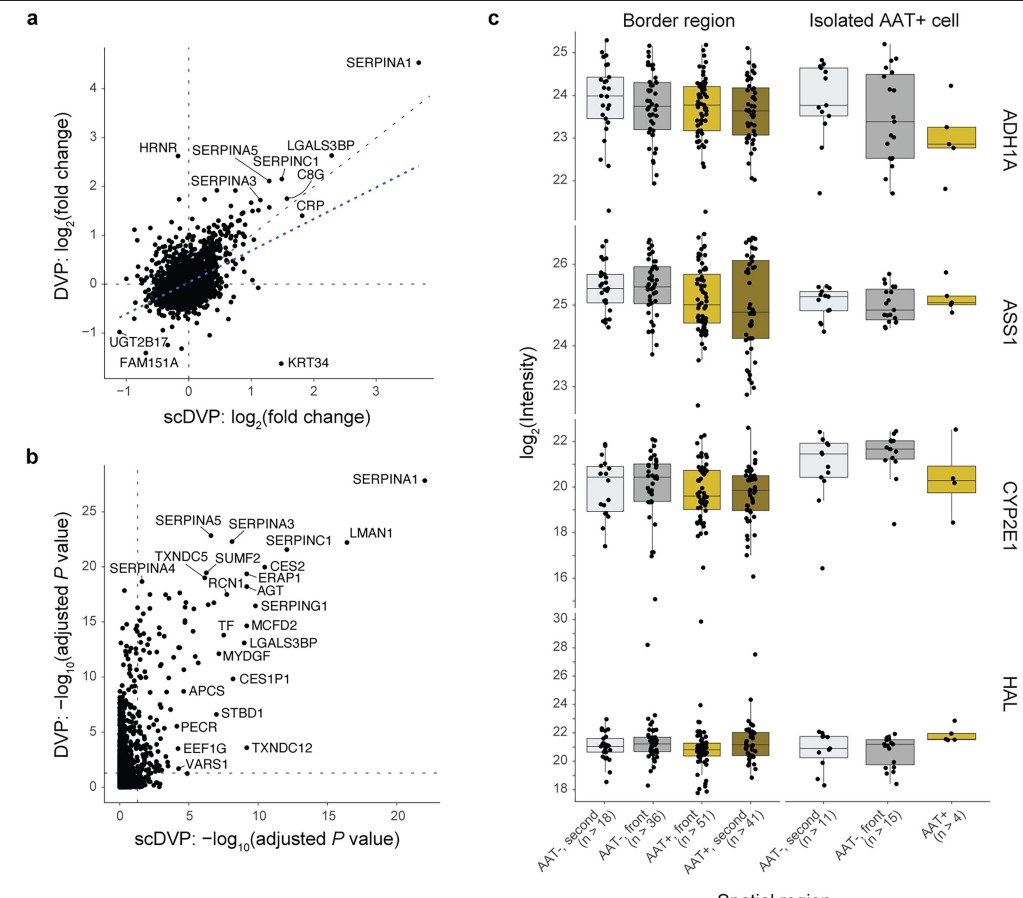

**Extended Data Fig. 7 | Comparison of the single-cell proteome with DVP class data and zonation. a**, Comparison of log₂(fold changes) and **b**, adjusted *P* values between AAT+ and AAT- single-cell comparisons (x-axis) versus cells along the accumulation gradient (y-axis; refer to Figs. 1 and 2). Statistics as in

Extended Data Fig. 6h, and Fig. 1c. **c**, Protein expression in spatial regions for indicated markers. Periportal markers: ASS1 and HAL; pericentral markers: ALDH1A1 and CYP2E1. Box plots show first and third quartiles (box), median (thick line), and whiskers (±1.5 interquartile range).

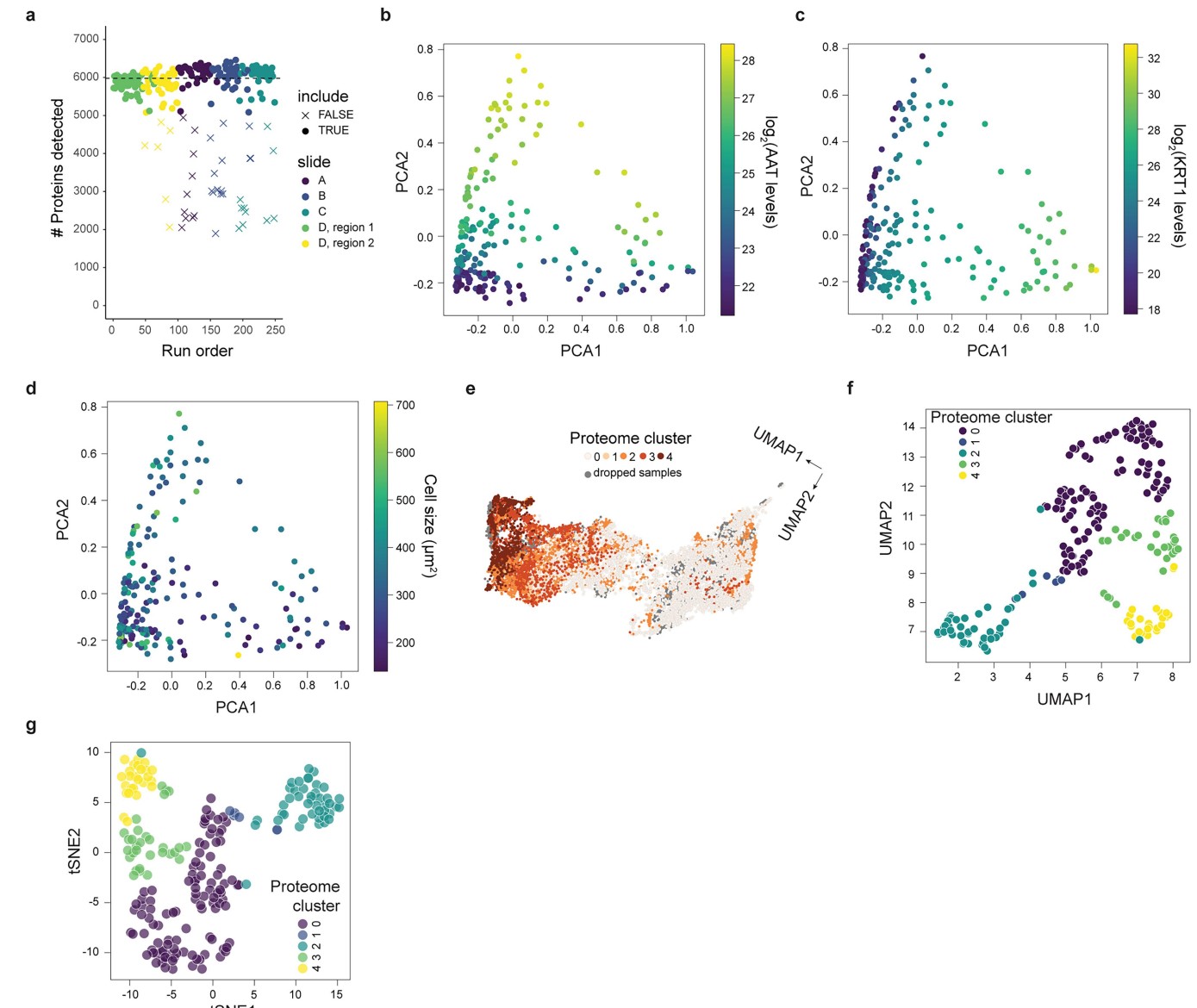

**Extended Data Fig. 8 | Quality control of morphology-guided DVP. a**, Number of protein groups detected per sample. Each dot is one sample, the horizontal line indicates the mean across all included samples (n = 209 included, n = 41 excluded and marked with a cross). Exclusion criteria were that the number of detected proteins was smaller than mean minus 0.5 SD. **b**, Principal component analysis of all included samples with AAT, **c**, KRT1 expression levels, or **d**, shape size color coded (n = 209). **e**, Annotation of the proteome cluster in Fig. 4d onto the image space UMAP. Dropped samples are in grey (n = 12,500). **f**, Representation of individual samples color coded by proteome cluster in a proteomic UMAP, or **g**, tSNE space (n = 209).

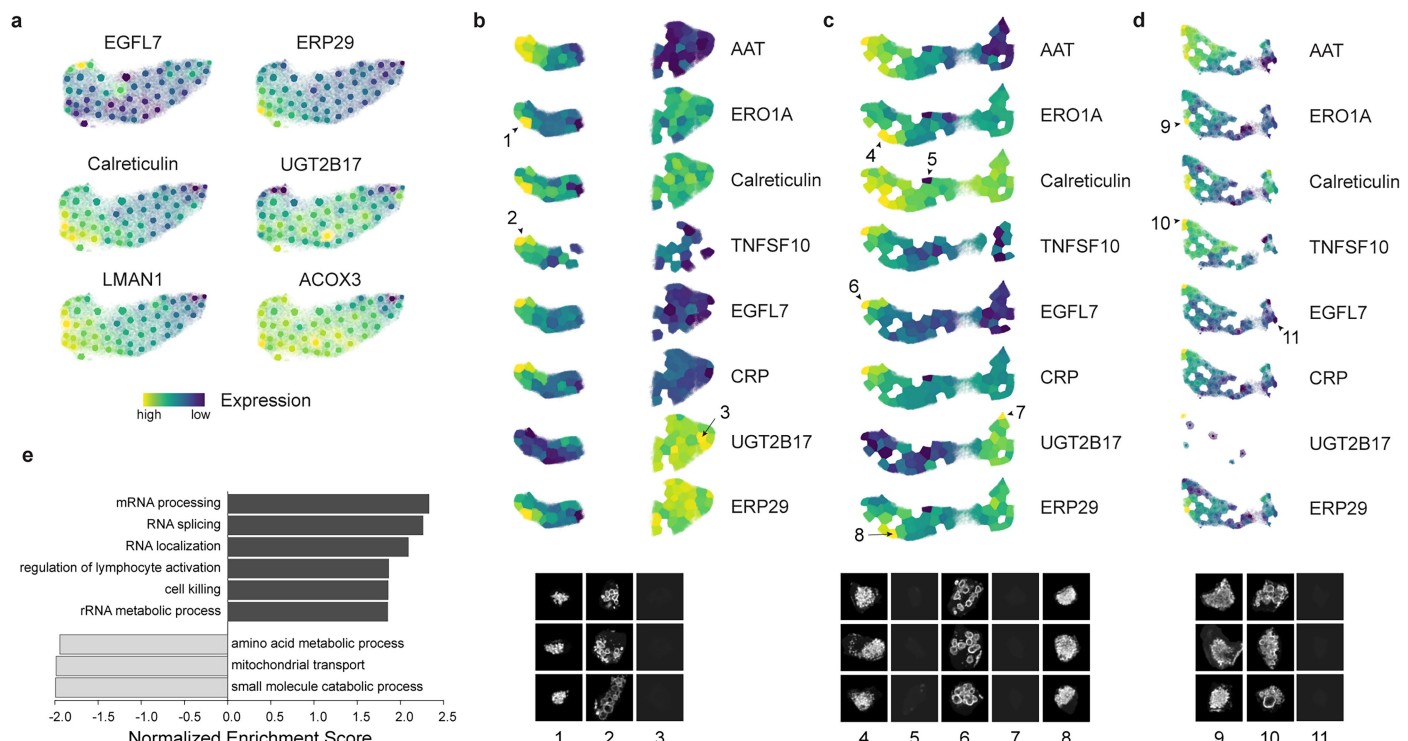

**Extended Data Fig. 9 | The proteome of cells with various aggregate morphologies. a-d**, Protein expression across phenotypic UMAP space. Each panel represents one tissue section (n = 4). Notable clusters indicated by arrows and numbers, with representative images shown below. **e**, Gene Set Enrichment Analysis (GO: Biological Process noRedundant) comparing globular versus amorphous aggregate types.

# Reporting Summary

## Statistics

For all statistical analyses, confirm that the following items are present in the figure legend, table legend, main text, or Methods section.

| n/a | Confirmed | |
|---|---|---|
| ☐ | ☒ | The exact sample size (*n*) for each experimental group/condition, given as a discrete number and unit of measurement |
| ☐ | ☒ | A statement on whether measurements were taken from distinct samples or whether the same sample was measured repeatedly |
| ☐ | ☒ | The statistical test(s) used AND whether they are one- or two-sided *Only common tests should be described solely by name; describe more complex techniques in the Methods section.* |
| ☐ | ☒ | A description of all covariates tested |
| ☐ | ☒ | A description of any assumptions or corrections, such as tests of normality and adjustment for multiple comparisons |
| ☐ | ☒ | A full description of the statistical parameters including central tendency (e.g. means) or other basic estimates (e.g. regression coefficient) AND variation (e.g. standard deviation) or associated estimates of uncertainty (e.g. confidence intervals) |
| ☐ | ☒ | For null hypothesis testing, the test statistic (e.g. *F*, *t*, *r*) with confidence intervals, effect sizes, degrees of freedom and *P* value noted *Give P values as exact values whenever suitable.* |
| ☐ | ☒ | For Bayesian analysis, information on the choice of priors and Markov chain Monte Carlo settings |
| ☐ | ☒ | For hierarchical and complex designs, identification of the appropriate level for tests and full reporting of outcomes |
| ☐ | ☒ | Estimates of effect sizes (e.g. Cohen's *d*, Pearson's *r*), indicating how they were calculated |

*Our web collection on statistics for biologists contains articles on many of the points above.*

## Software and code

Policy information about availability of computer code

| Data collection | Zeiss ZEN Blue (3.7.97.07000), PerkinElmer Harmony v4.9, Single-Cell Technologies BIAS (September 2023), Leica LMD Beta10, Orbitrap Astral Tune Application 1.0.100.40, Cellpose 2.0, scPortrait, ConvNext, huggingface v4.26 |
|---|---|
| Data analysis | DIA-NN v1.8.1, , directLFQ v0.2.19, R v4.4.1, limma v3.60.3, PCAtools v2.16.0, WebGestalt 2024, umap-learn 0.5.6, scikit-learn 1.4.2, pandas 2.2.1, numpy 1.26.4 |

For manuscripts utilizing custom algorithms or software that are central to the research but not yet described in published literature, software must be made available to editors and reviewers. We strongly encourage code deposition in a community repository (e.g. GitHub). See the Nature Portfolio guidelines for submitting code & software for further information.

# Data

Policy information about [availability of data](availability of data)

All manuscripts must include a [data availability statement](data availability statement). This statement should provide the following information, where applicable:

- Accession codes, unique identifiers, or web links for publicly available datasets
- A description of any restrictions on data availability
- For clinical datasets or third party data, please ensure that the statement adheres to our [policy](policy)

The mass spectrometry proteomics data have been deposited to the ProteomeExchange Consortium via the PRIDE partner repository with the dataset identifier PXD054440. The R and Python code used in this study is documented at https://github.com/MannLabs/Proteotoxicity. Imaging data of explant and morphological clusters has been deposited to BioStudies with the identifier S-BIAD1523.

# Research involving human participants, their data, or biological material

Policy information about studies with [human participants or human data](human participants or human data). See also policy information about [sex, gender (identity/presentation), and sexual orientation](sex, gender (identity/presentation), and sexual orientation) and [race, ethnicity and racism](race, ethnicity and racism).

| | |
|---|---|
| Reporting on sex and gender | Clinical metadata including sex are summarised in Extended Data Fig. S1A. Both male and female samples were included without prior selection due to the small cohort size. |
| Reporting on race, ethnicity, or other socially relevant groupings | Socially constructed categorization was not taken into consideration. Patient material was sampled in Odense, Denmark, and Aachen, Germany, and represent the patient cohort in the larger region. |
| Population characteristics | Clinical metadata including age, alcohol consumption, diabetes and BMI are summarised in Extended Data Fig. S1A. Individual samples were explicitly de-identified to to comply with European and country-specific General Data Protection Regulation (GDPR). |
| Recruitment | Odense patient recruitment – Patients were recruited through the Danish patient organization (Alfa-1 Denmark) and clinical departments for liver and lung diseases as part of a cohort study. The cohort was designed to investigate liver health among non-pregnant adults (minimum age 18 years) diagnosed with AATD of any genotype and carrier status. This specific study includes 16 individuals diagnosed with Pi*ZZ who consented to undergo the procedure. Participants without a history of liver transplant or decompensated cirrhosis were offered a percutaneous liver biopsy. The patients underwent liver core needle biopsies at Odense University Hospital (OUH) between 2017 and 2021. 

Aachen patient recruitment – The recruitment of patients is described in detail in reference 34. Of this cohort, the present study includes 19 individuals diagnosed with Pi*ZZ, of whom 14 underwent liver core needle biopsies due to medical indication and five received a liver transplantation due to end-stage liver disease. |
| Ethics oversight | Odense: The study was approved by the Danish Ethical Committee (S-20160187), and participants gave informed consent prior to enrollment. Aachen: thical approval was provided by the institutional review board of Aachen University (EK 173/15). All participants provided written informed consent and were treated following the ethical guidelines of the Helsinki Declaration (Hong Kong Amendment) as well as Good Clinical Practice (European guidelines). |

Note that full information on the approval of the study protocol must also be provided in the manuscript.

# Field-specific reporting

Please select the one below that is the best fit for your research. If you are not sure, read the appropriate sections before making your selection.

[x] Life sciences      [ ] Behavioural & social sciences      [ ] Ecological, evolutionary & environmental sciences

For a reference copy of the document with all sections, see [nature.com/documents/nr-reporting-summary-flat.pdf](nature.com/documents/nr-reporting-summary-flat.pdf)

# Life sciences study design

All studies must disclose on these points even when the disclosure is negative.

| | |
|---|---|
| Sample size | Thirty-four of 35 available patient Pi*ZZ samples were included. One sample was excluded due to its outlier position on a PCA. |
| Data exclusions | MS samples were included if the number of protein groups exceeded (a) the mean minus 1.5 standard deviations for DVP, resulting in 5.9% (6/102) dropouts; (b) the mean minus 0.5 standard deviations for DVP-ML samples; (c) a fitted logarithmic curve minus 1.5 interquartile ranges for scDVP, taking the relation between size and proteomic depth into account, resulting in 15.4% (40/259) dropouts. The lower cutoffs were selected after manual inspection of the data distribution. For morphology-guided DVP, one explant section was removed from analysis due to too few AAT-positive cells in the section. |
| Replication | Only one technical replicate was used and analysed per patient biopsy (see Supplementary Table S1). |

| | |
|---|---|
| Randomization | Patient samples were processed in random order. MS acquisition was not randomized and MS performance was tightly monitored with HeLa QC runs before, during, and after each experimental block. |
| Blinding | Cell selection was automated and driven by k means clustering or convolutional neural networks without human input. Investigators were blinded for laser microdissection. Data analysis was not blinded to ensure correct handling of biological positive controls. |

# Reporting for specific materials, systems and methods

We require information from authors about some types of materials, experimental systems and methods used in many studies. Here, indicate whether each material, system or method listed is relevant to your study. If you are not sure if a list item applies to your research, read the appropriate section before selecting a response.

## Materials & experimental systems

| n/a | Involved in the study |
|---|---|
| ☐ ☒ | Antibodies |
| ☒ ☐ | Eukaryotic cell lines |
| ☒ ☐ | Palaeontology and archaeology |
| ☒ ☐ | Animals and other organisms |
| ☐ ☒ | Clinical data |
| ☒ ☐ | Dual use research of concern |
| ☒ ☐ | Plants |

## Methods

| n/a | Involved in the study |
|---|---|
| ☒ ☐ | ChIP-seq |
| ☒ ☐ | Flow cytometry |
| ☒ ☐ | MRI-based neuroimaging |

## Antibodies

| | |
|---|---|
| Antibodies used | PRIMARY: Mouse IgG1 monoclonal AAT 2C1 (1:200, Hycult HM2289); Rabbit recombinant anti-pan cadherin [EPR1792Y] (1:200, Abcam ab51034)<br>SECONDARY: Goat anti-mouse IgG1 (1:400 Invitrogen A21127); Goat anti-rabbit AF647 (1:400, Invitrogen A21245) |
| Validation | The mouse anti-human AAT monoclonal antibody 2C1 was tested positive in an ELISA assay according to the manufacturer: https://www.hycultbiotech.com/product/alpha-1-antitrypsin-human-mab-2cl/. The rabbit monoclonal anti-pan cadherin antibody [EPR1792Y] was tested by the manufacturer Abcam (ab51034) in Western blot on recombinant protein, by flow cytometry against an isotype control, and through OI-RD scanning. |

## Clinical data

Policy information about clinical studies
All manuscripts should comply with the ICMJE guidelines for publication of clinical research and a completed CONSORT checklist must be included with all submissions.

| | |
|---|---|
| Clinical trial registration | N/A |
| Study protocol | N/A |
| Data collection | N/A |
| Outcomes | N/A |

## Plants

| | |
|---|---|
| Seed stocks | N/A |
| Novel plant genotypes | N/A |
| Authentication | N/A |

