## [Peer Review File · Nature]

Deep Visual Proteomics maps proteotoxicity in a genetic liver disease

Corresponding Author: Professor Matthias Mann

Version 0:

Reviewer comments:

Referee #1

(Remarks to the Author)

This is an interesting paper that takes a novel approach of integrating Deep Visual Proteomics (DVP) with single-cell analysis on liver sections from patients with antitrypsin deficiency. The paper is well written and the data are clear. The authors have curated a cohort of formalin-fixed paraffin-embedded (FFPE) biopsies and liver explants from 35 patients homozygous for the pathogenic Z-variant (Pi*ZZ), encompassing all fibrosis stages. The population were matched for BMI and age. When dividing into the 4 fibrosis sections (F1-F4) the numbers become small and so it is not possible to match for alcohol consumption which may be important in liver disease in antitrypsin deficiency (0/3 alcohol consumption in F3 group). I am confused about the number of subjects in the Table in Extended Data Fig. 1. Adding up numbers for the sex gives 32 subjects and alcohol 33 subjects (counting NA as subjects). The legend to Figure 1 reports 32 subjects.

Proteomic analysis following laser microdissection revealed 5,000 proteins/sample with cells divided into those with high, moderate and low load of antitrypsin (SERPIN A1). The authors comment 'Given the sparsity of AAT+ cells in biopsy material'. How were the areas selected for 3 µm micro dissection and can the authors be confident that selection by eye does not introduce bias? The results show enrichment of the chaperone HSPA5 and LMAN1. This increase in chaperones and XBP-1, ATF4 and ATF6 signalling is interesting and fits with existing data that there is evidence of ER stress in hepatocytes from individuals with antitrypsin deficiency but not in cell lines that express only the mutant protein. This supports the importance of a 'second hit' in antitrypsin deficiency liver disease whether it be fat or alcohol.

The authors write 'Among the most dysregulated hits, we identified other secretory proteins, including many SERPINS, coagulation, and complement factors' and 'This corroborates the notion of ineffective processing and crowding in the ER space, with pathological implications due to the systemic deficiency of multiple plasma proteins 16' However this is not apparent in clinical practice until there is advanced liver disease/liver failure at which stage synthetic failure is a feature of all forms of liver disease. Do these data provide insights into the deficiency of specific plasma proteins as result of antitrypsin accumulation in the ER? The tight tracking of antitrypsin with related serpins is interesting but how do the authors explain this finding? I am not aware of any biochemical evidence that the different serpins interact and are retained within hepatocytes or that other serpin deficiencies co-exist with antitrypsin deficiency.

The analysis of early and late responders assumes that severity of disease is linearly associated with the accumulation of antitrypsin. However as the authors point out this is not the case 'the maximum load remained fairly equal across all stages (Extended Data Fig. 1g)' (line 108). Thus is it appropriate to draw these conclusions on disease progression based on low, medium and high antitrypsin load within individual cells?

The hallmark of antitrypsin deficiency liver disease are the PAS positive inclusions that are now recognised to be polymers. Do the authors know whether it is the polymer load within a given cell or the response to the misfolded protein that is causing cell toxicity?

The spatial proteomics data are very interesting and an attempt to define cross talk between cells at the earliest stages of fibrosis. The authors show an absence of dedicated stress propagation between neighbouring cells. The authors point out that 'the presence of large patches of positive cells implies a propagation mechanism' (line 278). This is interesting in the context of b-amyloid and Alzheimer disease but there is no evidence of this to date in antitrypsin deficiency.

The authors suggest that the data provide potentially drugable targets but it is not clear if these are specific to antitrypsin deficiency or if similar pathways are activated in other forms of progressive liver disease. This is an important control but would represent a huge amount of extra work.

In summary, this is a very nice piece of work and the authors should be congratulated on the depth of the proteomic analysis. However I am not convinced that this is of sufficient impact and general interest for publication in Nature. Perhaps it would be more appropriate for one of the sister journals?

Referee #2

(Remarks to the Author)

From the reviewer's perspective, the major contribution of this manuscript is to demonstrate that single-cell mass spectrometry-based spatial proteomics analysis of FFPE tissues, combined with morphological images, becomes a powerful tool for analyze disease progression. This is particularly useful for diseases where effective models are not available. This type of analysis cannot be approached with other technologies such as nucleic acids-based sequencing or antibody-based protein measurements. It is applicable to the achieved FFPE tissue sections, thus opening a new approach for understanding disease pathogenesis. The weakness of this study is that the biomedical insights are not validated mechanistically, and the analysis is based on a relatively small cohort. That said, I am still supporting it as a strong candidate for Nature, since the unique insights it brings based on FFPE tissues cannot be obtained by other means. It enables disease pathology to be analyzed directly on archived tissue sections at single cell and omics level.

That said, I suggest the authors to address the following issues in the revision.

1, The DVP technology presented here is not novel since the authors have published previously. Here they increased proteomic depth by 50% and applied to FFPE samples. Can you provide more details of the technical improvements?

2, In several sections (line 95, line 158, line 172, line 210), the authors analyzed a certain number of FFPE samples. Can they explain the rationale behind the sample selection, and how did they determine the region of interest? Due to the intra-tissue heterogeneity and intra-patient heterogeneity, the authors should provide evidence regarding whether the samples are representative. For example, in line 172, they selected three F1-stage biopsies (please explain the stages of the disease explicitly). Why F1? Why only three? How big is the area of these FFPE sections? Why did they selected 132 single shapes? Since all the downstream proteomics analysis and reasoning are based on data from these three biopsies, can they explain why they are representative and the conclusion here could be potentially extended to more samples in more patients. Around line 185, they claimed that the data supports absence of dedicated stress propagation between neighboring cells, suggesting that proteotoxic stress is a cell-intrinsic response. Indeed, this is what the data tell, but considering that only three biopsy samples were analyzed, a reasonable limitation statement is recommended. There are many other similar claims like this throughout the manuscript.

3, How is this approach compared to spatial transcriptomics? I suggest to include some discussion of the pros and cons, complementarity in the Discussion, to help readers better understand the boundary of the method.

4, typo: line 167: then  the ?

Referee #3

(Remarks to the Author)

Summary of the key results

The authors have used formalin fixed archived human live tissue (biopsies and explants) from Pi*ZZ individuals to perform a proteomics study with unusually high spatial resolution. They have grouped cells by their degree of alpha1 antitrypsin (AAT) content and found increased levels of some proteins correlate with AAT, e.g. higher levels of endoplasmic reticulum chaperones and peroxisomal proteins appear track levels of AAT. Treating AAT accumulation as a marker of time, some proteins increased in abundance 'early' (smaller increases in AAT) and other appeared 'late' (ie correlated with higher AAT abundance). Increased levels of ER proteins was interpreted as showing higher levels of the unfolded protein response (UPR), which was a 'late' event (ie correlated with higher AAT levels). Thanks to the high level of spatial resolution, the authors were able to show AAT accumulation differed between cells in the same sample and changed abruptly between areas of low and higher AAT, rather than showing gradation. The did not confirm a zonal distribution of AAT accumulation, which has been claimed by others. While ERO1A increased as AAT increased, cells containing 'globules' of AAT contained less ERO1A than expected, which they referred to as as being 'decreased'. By contrast, CRP was enriched more than they had expected, which they interpreted as "a terminal phenotype preceding intrinsic apoptosis".

Originality and significance: if not novel, please include reference

This is a highly innovative study using technologies developed by these authors. The datasets will be of great interest to the AAT research field.

Data & methodology: validity of approach, quality of data, quality of presentation

I am not able to comment on the machine learning nor on the proteomics. The data presentation is of a high quality.

Appropriate use of statistics and treatment of uncertainties

Statistical analysis appears appropriate

Conclusions: robustness, validity, reliability

The data appear robust, though some observations may have been over-interpreted (vide infra)

Suggested improvements: experiments, data for possible revision

This is an excellent study but consideration of the following points might allow the authors to improve the work.

1. One of the striking observations is the increased expression of ER proteins tracking the cellular level of AAT. This is interpreted as 'activation of the UPR'. While UPR activation is the mechanism by which the cell matches ER capacity with its load, a high ER content is not the same as UPR activation. Can the authors note if proteins correlating with current increased UPR signalling (e.g. XBP1s and/or CHOP) are elevated in cells thought to be exhibiting 'UPR activation'. Whilst protein signatures might be annotated as "XBP1 activation" this often simply reflects prior ER expansion. If current UPR active signalling cannot be confirmed, perhaps the term might be substituted with 'ER expansion'.

2. It is interesting that cell populations AAT+ (high AAT) and AAT- (absent or low) are found in groups showing "sharp borders". This is reminiscent of a new study demonstrating that within the cirrhotic livers of Pi*ZZ individuals, clones of cells frequently acquire SERPINA1 deletions or C-terminal truncations incompatible with polymerisation. Do the proteomics data provide sufficient AAT protein coverage to determine if patches of cells lacking AAT are enriched for C-terminal truncations?

3. It is noted that CRP levels are elevated in cells exhibiting what the authors take to be 'a terminal phenotype'. Based on the current data presented, it is not proved that this is a terminal phenotype, so could the authors provide more evidence that these cells are activating death pathways? If not, perhaps rewording is required. The observation that CRP is elevated is nevertheless interesting and it would be interesting to know if this represents increased transcription or impaired ER exit. It is likely that larger complexes, such as CRP heximers, might become trapped by the solidification of Z-AAT that has recently been described in the ER. Do the current data suggest any bias in the nature of ER client proteins that accumulate in cells in this 'advanced' state, for example are members of larger complexes less likely to be secreted?

References: appropriate credit to previous work?

The referencing appears appropriate.

Clarity and context: lucidity of abstract/summary, appropriateness of abstract, introduction and conclusions

These sections of the manuscript are very well written

Version 1:

Reviewer comments:

Referee #1

(Remarks to the Author)

Many thanks for asking me to review the revised manuscript. The detailed response to reviewers was very helpful and should be published along with the manuscript if the paper is accepted. The authors have added 3 new samples and have amended the text to deal with my comments. I think this has greatly improved the manuscript. The depth of the single cell proteomic mapping (up to 4,300 protein/cell) is very impressive and moves forward our understanding of cellular mechanisms in antitrypsin deficiency. In particular that peroxisomal upregulation precedes the canonical unfolded protein response and that elevated TNFSF10/TRAIL occurs late in the disease. The dataset will be of great value to the community and like all good data raises new questions that will require further study: what is the mechanism of dysregulation of secretory serpins, whether the intracellular changes impact on plasma proteins, the 'chicken and egg' question of whether it is the polymers themselves or the response to the polymers that is causing cellular injury and whether or not there is propagation of ER stress. The clusters of AT+ cells suggests that this is the case but there is no evidence at the cellular level.

A few minor points:

1. Extended Data Fig. 2 is missing a legend for panels g and k. The final sentence describes panels i-m and n but I can't see panels l, m or n.
2. Extended Data Fig. 5 panel b 'Significant pathways are color-coded'. There is no colour in the panel.
3. Extended Data Fig. 6 c, 'Protein quantification across all hepatocyte shape runs (n = 259). Lower dotted line: median in batch A (2601 proteins); upper dotted line: median in batch B (3004 proteins)'. The dotted lines are hard to see.

Referee #2

(Remarks to the Author)

Thanks for the revisions. I think my comments have been reasonably addressed. Congratulations for the nice paper.

Referee #3

(Remarks to the Author)

Thank you addressing my comments thoroughly.

Congratulations on an impressive study.

Point-by-point answers to reviewers

The landscape of proteotoxicity in a liver disease by Deep Visual Proteomics (Rosenberger *et al.*, Manuscript ID 2024-08-16761A)

Summary of changes

We sincerely thank all three reviewers for their positive and insightful comments on our manuscript, and for the time they invested in making it stronger. We are pleased that the reviewers believe it is “*an excellent piece of work*” and “*highly innovative*” (Reviewer #3) that is “*well written*” with “*clear data*” (Reviewer #1) and they support it “*as a strong candidate in Nature*” (Reviewer #2). We greatly appreciate their recognition of both the novelty and importance of applying Deep Visual Proteomics (DVP) to understand disease progression in Alpha-1 Antitrypsin Deficiency (AATD).

Their feedback has helped us to refine our work and to address areas requiring additional analysis, clarification or explanation. In our revised manuscript, we have incorporated all reviewers' comments and made specific changes to improve the robustness of our conclusions and data presentation. We provide detailed responses to each reviewer's points below. The most important changes are:

- We have added three more samples to our single-cell tissue mapping at AAT+/AAT- border regions, now totaling biopsies of six individuals, leading to more powerful statistics. Our new measurements confirm the previous findings. We also increased the number of detected proteins in a single hepatocyte slice to up to a remarkable number of 4,300 proteins. We find strong correlation between single cell DVP (scDVP) and regular DVP measurements (100 combined shapes) highlighting again that scDVP is biologically highly reliable.
- The reviewer comments point out the great resource value of this data. To make data access easier for the scientific communities, we updated the Supplementary Data Tables that now include binned data for more continuous alpha-1 antitrypsin analysis than the three categories before.
- Along these lines, we also established a comprehensive Github repository containing all R and Python code (Figure R1). Our repository is structured with a single master script in the RScripts folder that reproduces all manuscript figures locally, facilitating data access and further analysis. The reviewer access code is provided in the ‘Data Availability Section’.

Florian Rosenberger et al., 2024 - in revision

The proteomic landscape of proteotoxic stress in a fibrogenic liver disease

Abstract

Protein misfolding diseases, including alpha-1 antitrypsin deficiency (AATD), pose significant health challenges, with their cellular progression still poorly understood^{1–3}. We utilize spatial proteomics by mass spectrometry and machine learning to map AATD in human liver tissue. Combining Deep Visual Proteomics (DVP) with single-cell analysis^{4,5}, we probe intact patient biopsies to resolve molecular events during hepatocyte stress in pseudo-time across fibrosis stages. We achieve unprecedented proteome depth of up to 3,800 proteins from a third of a single cell in formalin-fixed, paraffin-embedded (FFPE) tissue. This dataset revealed a potentially clinically actionable peroxisomal upregulation that precedes the canonical unfolded protein response. Our single-cell proteomics data show alpha-1 antitrypsin accumulation is largely cell-intrinsic, with minimal stress propagation between hepatocytes. We integrated proteomic data with AI-guided image-based phenotyping across multiple disease

Figure R1: All R code that was used in the study is now documented on Github and can easily be interfaced locally.

Referees' comments:

Referee #1 (Remarks to the Author):

This is an interesting paper that takes a novel approach of integrating Deep Visual Proteomics (DVP) with single-cell analysis on liver sections from patients with antitrypsin deficiency. The paper is well written and the data are clear.

We thank reviewer #1 for the positive assessment of our work and the recognition of its strengths, particularly in terms of integrating proteomics with single-cell analysis.

The authors have curated a cohort of formalin-fixed paraffin-embedded (FFPE) biopsies and liver explants from 35 patients homozygous for the pathogenic Z-variant (Pi*ZZ), encompassing all fibrosis stages. The population were matched for BMI and age. When dividing into the 4 fibrosis sections (F1-F4) the numbers become small and so it is not possible to match for alcohol consumption which may be important in liver disease in antitrypsin deficiency (0/3 alcohol consumption in F3 group).

We agree that a larger study size might have allowed deeper insights into additional co-variables such as alcohol consumption. While such a larger cohort would be ideal, there are practical obstacles to this, while we note several factors that support the robustness of our current dataset:

1. **Sample Rarity:** Research-accessible fine needle aspirates from AATD patients, particularly those with low-grade fibrotic liver tissue, are exceptionally rare. Our cohort of 35 patients contains all early-stage fibrotic samples we were able to get access to, and retains a good balance with late-stage fibrotic sample numbers.
2. **Statistical Power:** Despite the limited samples per fibrosis stage, our key findings show strong statistical significance. The molecular signatures we identify are consistently observed across samples within each stage, demonstrating that the differences reflect true biological variation rather than technical noise. Furthermore, in sample comparisons by paired testing diminish potential cohort biases.
3. **Alcohol Consumption:** Unfortunately, matching for alcohol consumption across fibrosis stages was not possible due to sample availability (0/3 alcohol consumption in F3 group). Furthermore, our proteomic analysis focuses primarily on cell-intrinsic responses to AAT accumulation, which should be mostly independent of this variable.

To address this concern, we have now included a **limitation statement in the discussion** to accommodate the reviewer's comment:

"A limitation of this study is the low sample numbers due to limited availability of particularly low-grade fibrotic tissue. This prevents us from further disentangling confounding factors such as alcohol consumption. Nevertheless, the cellular enrichment by DVP allows the biological phenotype to emerge more clearly, leading to statistically robust and actionable insights even at low sample numbers." (new manuscript lines II. 264 – 269)

I am confused about the number of subjects in the Table in Extended Data Fig. 1. Adding up numbers for the sex gives 32 subjects and alcohol 33 subjects (counting NA as subjects). The legend to Figure 1 reports 32 subjects.

Thank you for pointing this out. We have now corrected and enlarged the **Extended Data Fig. 1a, Fig. 1a** and the figure legend. We have 35 patient samples included in total, of which we lack Kleiner scores of three samples (resulting in n = 32 for some analyses with Kleiner scores as a covariable). This number is also consistent with the Materials & Methods section.

Proteomic analysis following laser microdissection revealed 5,000 proteins/sample with cells divided into those with high, moderate and low load of antitrypsin (SERPIN A1). The authors comment 'Given the sparsity of AAT+ cells in biopsy material'. How were the areas selected for 3 µm micro dissection and can the authors be confident that selection by eye does not introduce bias?

Thank you for this comment, which allows us to clarify this important point. To strictly exclude human bias, we had isolated cells from the entire fine needle aspirates or tissue resections without focussing on one particular region. To achieve fair representation in low/middle/high aggregate load classes, we then applied a multilayer perceptron (MLP) classification approach based on cells' maximum, mean and median AAT staining intensity within the segmentation mask. Of these, we picked bin 1, 3 and 5, and termed those low/middle/high. To make our approach more clear, we have added the following to the methods section:

"(...) all cells per biopsy or explant tissue were divided into five classes using a multilayer perceptron (...)" (II. 397 - 399)

"Outlines of all cells per biopsy were identified in an unbiased way by using CellPose" (I. 392 - 393)

"Classification was based on the AAT (alpha-1 antitrypsin) maximum, median, and mean intensity within the cell outline mask, involving no human intervention." (II. 400 - 402)

In contrast to that, regions for single-cell spatial mapping in 10µm sections were selected manually based on presence of clear borders between negative and positive cells. All cells indicated in **Fig. 3d** and **Extended Data Fig. 3e/f** were isolated for proteome quantification, eliminating potential cell selection bias.

The results show enrichment of the chaperone HSPA5 and LMAN1. This increase in chaperones and XBP-1, ATF4 and ATF6 signalling is interesting and fits with existing data that there is evidence of ER stress in hepatocytes from individuals with antitrypsin deficiency but not in cell lines that express only the mutant protein. This supports the importance of a 'second hit' in antitrypsin deficiency liver disease whether it be fat or alcohol. The authors write 'Among the most dysregulated hits, we identified other secretory proteins, including many SERPINs, coagulation, and complement factors' and 'This corroborates the notion of

ineffective processing and crowding in the ER space, with pathological implications due to the systemic deficiency of multiple plasma proteins 16' However this is not apparent in clinical practice until there is advanced liver disease/liver failure at which stage synthetic failure is a feature of all forms of liver disease. Do these data provide insights into the deficiency of specific plasma proteins as result of antitrypsin accumulation in the ER?

The reviewer raises a clinically highly important question. To systematically map the effect of proteins that are produced in hepatocytes and released into blood, we have now integrated the new **Extended Data Fig. 2e**. Our data in fact shows significant accumulation of at least 72% of plasma-targeted proteins. We do not know whether this reflects in measurable changes in plasma and we do not have matched plasma samples from enough patients to measure this. In order to reflect this limitation, we have changed the text indicated by the reviewer to:

“(...) with potential systemic pathological implications due to accumulation of annotated plasma proteins in affected hepatocytes (Extended Data Fig. 2e).” (II. 117 - 119)

The tight tracking of antitrypsin with related serpins is interesting but how do the authors explain this finding? I am not aware of any biochemical evidence that the different serpins interact and are retained within hepatocytes or that other serpin deficiencies co-exist with antitrypsin deficiency.

The co-accumulation of other serpins with AAT is indeed interesting and not obvious. We propose three potential mechanisms that could explain this finding:

1. Physical crowding effects: Recent work by Chambers et al. (Science Advances, 2022) ¹ demonstrated that Z-AAT undergoes phase transition to a solid state in the ER. This physical transformation could create a molecular sieve effect, particularly affecting proteins of similar size and biophysical properties.
2. Shared trafficking machinery: SERPINs use common ER-to-Golgi trafficking pathways, notably the LMAN1-MCFD2 cargo receptor complex which we find significantly upregulated in AAT+ cells. Recent work by Zhang et al. (Biochem J, 2022) ² has shown that this complex is critical for AAT secretion. The accumulation of misfolded AAT could saturate this machinery and cause retention of other SERPINs that depend on the same transport system.
3. Structural similarity: Most SERPINs share a highly conserved fold consisting of 3 β -sheets and 8-9 α -helices ³. Given that the Z-mutation (E342K) disrupts the structural stability of AAT, it is possible that the cellular stress induced by AAT accumulation particularly affects the folding efficiency of proteins with similar structural complexity.

While we are somewhat limited in space to discuss all hypotheses in detail, the reviewer's question prompted us to include the following sentence in the main text:

“This aligns with recent findings of SERPIN sequestration in AAT-inclusions, and supports the notion of crowding in the ER space (...)”^{1,4} (II. 116 - 117)

Note that our data shows the accumulation of other SERPINs together with antitrypsin on the protein and peptide level. This excludes that shared identical peptide sequences across SERPINs lead to mis-quantification. A new panel showing this is now part of the **Extended Data Fig. 2d**, and we have added “unambiguous” to the main text (**I. 115**).

The analysis of early and late responders assumes that severity of disease is linearly associated with the accumulation of antitrypsin. However as the authors point out this is not the case ‘the maximum load remained fairly equal across all stages (Extended Data Fig. 1g)’ (line 108). Thus is it appropriate to draw these conclusions on disease progression based on low, medium and high antitrypsin load within individual cells?

This conclusion is indeed appropriate because our analysis takes two complementary approaches to understand disease progression: (a) Classification of responses based on AAT load categories (Fig. 1), and (b) Direct correlation of protein levels with AAT quantity across all samples (Fig. 2). This dual approach resolves the problem that disease severity is not linearly associated with the accumulation of antitrypsin by turning a categorical comparison into a continuous analysis.

Based on the reviewer’s comment, we have now further analyzed the original image dataset to understand whether the maximum AAT load increases with fibrosis stage. Image data from almost 3 million cells corroborates the finding that the maximum load does not change, but that the basal AAT load is increased in cells in highly fibrotic tissue samples (**Extended Data Fig. 1h**). This suggests that disease progression may be driven by the cumulative burden of stressed cells rather than increasing severity within individual cells.

Concerning the total AAT load increase with higher fibrosis stage, this has also been described previously by Clark *et al.* (2018) ⁵, which we have now referenced in the main text as:

“Biopsies with a low fibrosis stage exhibited lower AAT baseline loading compared to high fibrosis stages on both proteomics and imaging data in line with previous findings⁵(...)” (II. 108)

The hallmark of antitrypsin deficiency liver disease are the PAS positive inclusions that are now recognised to be polymers. Do the authors know whether it is the polymer load within a given cell or the response to the misfolded protein that is causing cell toxicity?

This is a fundamental “chicken and egg” question about AATD pathology that is not directly addressed in our study. Nevertheless, our data provides several insights into this relationship between polymer load and cellular responses:

1. Our proteomic analysis shows that early responses (including peroxisomal activation and LGALS3BP upregulation) occur at relatively low AAT accumulation levels, before the dramatic expansion of ER stress markers. This suggests that initial cellular responses are triggered by the presence of misfolded protein rather than high polymer load alone. Of note, the very early markers are not altered in first- and second line hepatocytes at border regions in the single-cell spatial mapping dataset, corroborating the notion of absence of stress propagation (**Figure 3c**).

2. However, in cells with globular aggregates (which likely represent concentrated polymers based on their morphology), we observe a distinct proteomic signature including decreased UPR markers and elevated TNFSF10/TRAIL expression, suggesting a terminal state. This indicates that high polymer load may ultimately overwhelm cellular adaptation mechanisms.

The sequential nature of our observations - from early adaptive responses to a terminal phenotype - suggests that both mechanisms contribute to disease progression, with initial cellular responses to misfolded protein eventually being overwhelmed by increasing polymer load. This model aligns with recent work by Chambers et al. (Science Advances, 2022) ¹ showing that Z-AAT undergoes phase transition to a solid state in the ER, potentially explaining how polymer accumulation could ultimately disrupt cellular homeostasis. However, we feel that these observations and explanations are somewhat too speculative to merit detailed discussion in our manuscript.

The spatial proteomics data are very interesting and an attempt to define cross talk between cells at the earliest stages of fibrosis. The authors show an absence of dedicated stress propagation between neighbouring cells. The authors point out that 'the presence of large patches of positive cells implies a propagation mechanism' (line 278). This is interesting in the context of b-amyloid and Alzheimer disease but there is no evidence of this to date in antitrypsin deficiency. The authors suggest that the data provide potentially drugable targets but it is not clear if these are specific to antitrypsin deficiency or if similar pathways are activated in other forms of progressive liver disease. This is an important control but would represent a huge amount of extra work.

We agree with the reviewer that our data cannot directly answer whether potential druggable targets or pathways are specific to AATD. We also agree that including experimental evidence with DVP on various liver diseases would be beyond the scope of a single manuscript. However, prompted by the reviewer we turned to bulk proteomic datasets across fibrotic stages in our previously published dataset on alcohol-related liver disease (ALD) by Niu *et al.* (Nat Medicine, 2022) ⁶. Strikingly, the peroxisomal compartment that we found as a potentially druggable pathway in AATD is not significantly related to Kleiner scores in our ALD study, suggesting that at least this mechanism is specifically important in AATD. This observation is now included in the discussion:

"Of note, a peroxisomal response is not significantly correlated with fibrotic stages in bulk liver proteomes of alcohol-related liver disease (ALD) patients, suggesting that it is specifically important to the AATD-pathomechanism ⁶." (II. 313 - 316)

Along these lines, it is arguably not as relevant for tackling AATD or other proteotoxic diseases whether the targets are specific to a particular disease. If such a drug was clinically actionable in multiple settings it could still have enormous benefits.

In summary, this is a very nice piece of work and the authors should be congratulated on the depth of the proteomic analysis. However I am not convinced that this is of sufficient impact and general interest for publication in Nature. Perhaps it would be more appropriate for one of the sister journals?

We thank the reviewer once again for their important comments. We agree that the strength of the manuscript lies in its interdisciplinary and methodologically novel approach to important questions that previously could not be addressed. This includes the unprecedented biological content of single-cell proteomic mappings to a depth of up to 4,300 proteins, the CNN-guided stratification of aggregate morphologies, and novel biological insights into proteotoxicity with spatial dimension and along a pseudo-time axis. Furthermore, we have put considerable effort into making the data accessible to the scientific communities. Altogether, we rather agree with the other reviewers that this work is indeed a very strong candidate for *Nature* and its interdisciplinary audience.

Referee #2 (Remarks to the Author):

From the reviewer's perspective, the major contribution of this manuscript is to demonstrate that single-cell mass spectrometry-based spatial proteomics analysis of FFPE tissues, combined with morphological images, becomes a powerful tool for analyze disease progression. This is particularly useful for diseases where effective models are not available. This type of analysis cannot be approached with other technologies such as nucleic acids-based sequencing or antibody-based protein measurements. It is applicable to the achieved FFPE tissue sections, thus opening a new approach for understanding disease pathogenesis. The weakness of this study is that the biomedical insights are not validated mechanistically, and the analysis is based on a relatively small cohort. That said, I am still supporting it as a strong candidate for Nature, since the unique insights it brings based on FFPE tissues cannot be obtained by other means. It enables disease pathology to be analyzed directly on archived tissue sections at single cell and omics level.

We highly appreciate reviewer #2's recognition of our approach's power in analyzing archived FFPE tissue sections at the single-cell level, and its potential for advancing the understanding of diseases like alpha-1 antitrypsin deficiency. This experimental capability is especially relevant for conditions where tissue samples are rare, such as early fibrotic liver disease samples from AATD patients. We believe that additional molecular validation would shift focus from the proteomic landmark study that this manuscript represents. To accurately reflect this, we suggest changing the title from:

"The proteomic landscape of proteotoxic stress in a fibrogenic liver disease"

to

"The landscape of proteotoxicity in a liver disease by Deep Visual Proteomics".

That said, I suggest the authors to address the following issues in the revision.

1, The DVP technology presented here is not novel since the authors have published previously. Here they increased proteomic depth by 50% and applied to FFPE samples. Can you provide more details of the technical improvements?

Our manuscript starts with DVP of different load classes of cells with variable alpha-1 antitrypsin inclusion levels. While this builds on established pipelines, notably the base workflow by Mund et al. (2022) ⁷ and a modified antigen-retrieval method suitable for laser-

microdissection compatible membrane slides ⁸, we achieve much higher sensitivity than in the original paper (only 100 shapes vs. 700). This reflects an accumulation of incremental improvements throughout our DVP workflow published two years ago that nevertheless opens entirely new application areas.

Apart from this, conceptual novelties include:

(1) Single-cell DVP (scDVP) on single hepatocytes. Building on our previous work (Rosenberger et al. 2023) ⁹, we implemented several important technological advances:

- Application of the new Thermo Scientific Orbitrap Astral mass spectrometer for spatial tissue mapping by mass spectrometry.
- Implementation of a novel variable-window DIA mode optimizing MS2 sampling over the MS1 m/z precursor distribution increasing proteomics depth (see below).
- Demonstration of single-cell DVP on FFPE tissue, showing broader clinical applicability compared to our previous frozen section analyses ⁶. Despite the challenges of proteomic analysis on formalin-fixed material requiring antigen retrieval, we achieved unprecedented proteomic depth and biological insight.

(2) CNN-DVP on hepatocyte groups (50 cells each) using CNN-based morphological clustering of AAT patterns. This morphology guided spatial proteomics allows the direct functional investigation of cells with specific morphological features which turned out to be crucial to distinguish the biological paths taken by different cells in response to the proteotoxicity. To our knowledge, nothing similar exists in the single cell world. While predicted in the original DVP paper by Mund et al., this approach had not been previously achieved. We believe this will spark interest across biological disciplines where distinct morphologies correlate with specific functions.

To highlight these technological advances while maintaining accessibility to non-MS specialists, we have now updated the abstract, scDVP main text and discussion (**II. 274 - 275**) to reference the new instrument type and variable window design, also now highlighted in **Extended Data Figure S6A**. The discussion already contains a section on the novelty of the morphology-guided DVP approach (**II. 296 - 302**):

“We present an integration of image featurization and DVP that enables characterization of the entire proteomic and phenotypic lifecycle of stressed hepatocytes in a proteotoxic and fibrogenic liver disease. This methodology establishes a robust framework for dissecting complex cellular processes in situ across a spectrum of proteotoxic diseases. This strategy, an example of digital pathology with quantitative and very deep proteomic readout, yielded exceptionally deep proteomes of 6,000 quantified proteins, sufficient to infer most of the functional proteome of a given cell type.”

2, In several sections (line 95, line 158, line 172, line 210), the authors analyzed a certain number of FFPE samples. Can they explain the rationale behind the sample selection, and how did they determine the region of interest?

The full cohort used in the DVP experiment was determined by sample availability at clinical sites. We note that tissue samples, especially from early fibrotic stages, are extremely rare in Europe. In this revision, we have now expanded the scDVP sample size to biopsies from six individuals, which provides appropriate statistical power given the consistency of results and the early-stage nature of the technology as shown by our results. For explant sections, we initially selected five samples to maintain adequate MS analysis throughput. One sample was subsequently excluded from data analysis due to insufficient AAT+ cells for meaningful morphological classification. We have added methodological considerations to the scDVP section (II. 172 - 174).

Regarding region selection, for DVP experiments we analyzed cells from entire fine needle aspirates or tissue resections without regional bias (see also response to reviewer 1). To ensure fair representation across low/middle/high aggregate load classes, we implemented a multilayer perceptron (MLP) classification approach based on maximum, mean, and median AAT staining intensity of the cells within the segmentation mask, as now specified in the methods section (II. 397 - 399). For scDVP experiments, regions of interest were manually selected to analyze areas with directly adjacent AAT+ and AAT- cells (as stated in I. 407 - 411). For morphology-guided DVP analysis, we sampled from all nodules containing AAT+ cells within a tissue region.

Due to the intra-tissue heterogeneity and intra-patient heterogeneity, the authors should provide evidence regarding whether the samples are representative. For example, in line 172, they selected three F1-stage biopsies (please explain the stages of the disease explicitly). Why F1? Why only three? How big is the area of these FFPE sections? Why did they selected 132 single shapes? Since all the downstream proteomics analysis and reasoning are based on data from these three biopsies, can they explain why they are representative and the conclusion here could be potentially extended to more samples in more patients. Around line 185, they claimed that the data supports absence of dedicated stress propagation between neighboring cells, suggesting that proteotoxic stress is a cell-intrinsic response. Indeed, this is what the data tell, but considering that only three biopsy samples were analyzed, a reasonable limitation statement is recommended. There are many other similar claims like this throughout the manuscript.

We thank the reviewer for this opportunity to expand on our experimental setup. We have now increased our sample size to six F1-stage biopsies. We specifically selected F1 samples to understand aggregation mechanisms in a largely intact tissue environment and to minimize spatial confounders, such as altered blood flow from fibrotic scarring and elevated presence of fibrogenic hepatic stellate cells. We analyzed all available F1 samples that exhibited distinct AAT+/AAT- borders, reaching the maximum available number of biopsies of $n = 6$, as now specified in the main text (II. 172 - 174). The number of single-cell shapes was increased to 259 (previously 132) based on practical considerations related to the 384-well plate layout, which accommodates 22 shapes per row for convenient preparation and loading onto Evotips, thus minimizing potential batch effects and human error.

We analyzed the additional samples at single-cell resolution over a larger area (now specified in **Extended Data Figure S6D**) and they reproduce our key biological findings. These include the early stress signatures marked by LGALS3BP upregulation in cells with amorphous aggregates, and late stress signatures characterized by DNAJB11 and TNFSF10 upregulation in cells with globular aggregates. Importantly, we consistently observe that adjacent AAT+ or AAT- cells show highly similar proteomes, supporting our conclusion that SERPINA1 aggregate fragments do not trigger an early response in neighboring AAT- cells.

In addition, we have added a limitation statement to the discussion as mentioned already in response to reviewer 1:

“A limitation of this study is the low sample numbers due to limited availability of particularly low-grade fibrotic tissue. This prevents us from further disentangling confounding factors such as alcohol consumption. Nevertheless, the cellular enrichment by DVP allows the biological phenotype to emerge more clearly, leading to statistically robust and actionable insights even at low sample numbers.” (II. 264 – 269)

3, How is this approach compared to spatial transcriptomics? I suggest to include some discussion of the pros and cons, complementarity in the Discussion, to help readers better understand the boundary of the method.

We thank reviewer #2 for suggesting that we discuss DVP in comparison to spatial transcriptomics. We have now included the following statement in the discussion (**II. 278 - 286**):

“Spatial transcriptomics has become a powerful tool for spatial analyses in intact FFPE tissue, often approaching single cell resolution ¹⁰. In contrast, the scDVP approach provides orthogonal biological insights by directly measuring protein abundance with single cell localization. This is particularly valuable when post-transcriptional regulation and protein accumulation are central to pathology, such as for understanding proteotoxic diseases ¹⁰. While the scDVP approach is currently limited in throughput compared to transcriptomics, its combination with the herein presented morphology-guided DVP allows to sample efficiently over histologically heterogenous. This could be expanded into morphology-based proteome prediction for large number of cells.”

4, typo: line 167: then  the ?

We fixed this. – Thank you again for your valuable comments that make the manuscript more useful for the scientific communities.

Referee #3 (Remarks to the Author):

Summary of the key results

The authors have used formalin fixed archived human live tissue (biopsies and explants) from Pi*ZZ individuals to perform a proteomics study with unusually high spatial resolution. They have grouped cells by their degree of alpha1 antitrypsin (AAT) content and found increased levels of some proteins correlate with AAT, e.g. higher levels of endoplasmic reticulum

chaperones and peroxisomal proteins appear track levels of AAT. Treating AAT accumulation as a marker of time, some proteins increased in abundance 'early' (smaller increases in AAT) and other appeared 'late' (ie correlated with higher AAT abundance). Increased levels of ER proteins was interpreted as showing higher levels of the unfolded protein response (UPR), which was a 'late' event (ie correlated with higher AAT levels). Thanks to the high level of spatial resolution, the authors were able to show AAT accumulation differed between cells in the same sample and changed abruptly between areas of low and higher AAT, rather than showing gradation. The did not confirm a zonal distribution of AAT accumulation, which has been claimed by others. While ERO1A increased as AAT increased, cells containing 'globules' of AAT contained less ERO1A than expected, which they referred to as as being 'decreased'. By contrast, CRP was enriched more than they had expected, which they interpreted as "a terminal phenotype preceding intrinsic apoptosis".

Originality and significance: if not novel, please include reference

This is a highly innovative study using technologies developed by these authors. The datasets will be of great interest to the AAT research field.

Data & methodology: validity of approach, quality of data, quality of presentation

I am not able to comment on the machine learning nor on the proteomics. The data presentation is of a high quality.

Appropriate use of statistics and treatment of uncertainties

Statistical analysis appears appropriate

Conclusions: robustness, validity, reliability

The data appear robust, though some observations may have been over-interpreted (vide infra)

We are grateful for the reviewer's positive evaluation of our study and recognition of its innovative character and high quality of the data. We have carefully considered the reviewer's comments on the interpretation of certain findings and made revisions to clarify our conclusions as outlined below.

Suggested improvements: experiments, data for possible revision

This is an excellent study but consideration of the following points might allow the authors to improve the work.

1. One of the striking observations is the increased expression of ER proteins tracking the cellular level of AAT. This is interpreted as 'activation of the UPR'. While UPR activation is the mechanism by which the cell matches ER capacity with its load, a high ER content is not the same as UPR activation. Can the authors note if proteins correlating with `_current_` increased UPR signalling (e.g. XBP1s and/or CHOP) are elevated in cells thought to be exhibiting 'UPR activation'. Whilst protein signatures might be annotated as "XBP1 activation" this often simply

reflects prior ER expansion. If current UPR active signalling cannot be confirmed, perhaps the term might be substituted with 'ER expansion'.

We thank the reviewer for this valuable point. While we could not directly detect the levels of XBP1, CHOP or ATF6 in our proteomic analysis, many important UPR effectors in all three branches were highly elevated (**Extended Data Fig. 3e and 3f**). To distinguish between UPR activation and general ER expansion, we performed a comparative analysis of UPR-regulated versus non-UPR ER-resident proteins, based on literature-curated gene sets. This analysis shows that the UPR-related proteins are much more highly expressed than baseline levels (**Figure R2**), which is now also part of the manuscript as **Extended Data Fig. S2f**. In our view, this justifies the term 'UPR activation'.

Figure R2, Key proteins of the unfolded protein response ($x = TRUE$) in relation to other ER proteins ($x = FALSE$).

2. It is interesting that cell populations AAT+ (high AAT) and AAT- (absent or low) are found in groups showing "sharp borders". This is reminiscent of a new study demonstrating that within the cirrhotic livers of Pi*ZZ individuals, clones of cells frequently acquire SERPINA1 deletions or C-terminal truncations incompatible with polymerisation. Do the proteomics data provide sufficient AAT protein coverage to determine if patches of cells lacking AAT are enriched for C-terminal truncations?

This is an intriguing question. We are aware of the recent study by Brzowska *et al.*¹¹ demonstrating clonal expansions of cells with C-terminal SERPINA1 truncations. Their Figure 3 shows that nonsense mutations leading to sample-specific stop-codons occur in the region following the Pi*Z variant (E366K). Four tryptic peptides in the sequence could potentially detect such C-terminal truncations (**Figure R3**). While peptides #37, #40, and #41 showed poor coverage, peptide #39 was detected in 74.3% of AAT-negative shapes (as defined by immunofluorescence) in our single-shape dataset. We compared the intensity of peptide #39 with the well-covered N-terminal peptide #3, analyzing each biopsy separately to account for sample-specific genetics. However, we found no evidence of C-terminal truncations in any sample. Although the proteomics data for peptide #39 is robust, our sample size may currently be insufficient to detect such genetic events.

Figure R3, Top – tryptic peptides arising from the C-terminal end of AAT, and number of times they were detected in our scDVP experiment. Bottom – Ratio of peptide #3 (N-terminal end) to Peptide #39 (C-terminal end), stratified by scDVP sample to take sample-specific truncation events into account.

It is noted that CRP levels are elevated in cells exhibiting what the authors take to be 'a terminal phenotype'. Based on the current data presented, it is not proved that this is a terminal phenotype, so could the authors provide more evidence that these cells are activating death pathways? If not, perhaps rewording is required. The observation that CRP is elevated is nevertheless interesting and it would be interesting to know if this represents increased transcription or impaired ER exit.

We thank the reviewer for this comment. Analysis of our data does not indicate activation of cell death pathways in the analyzed cells based on two approaches: (1) apoptosis-related protein levels remained largely unchanged, and (2) targeted search for non-tryptic peptides from cleaved caspases were not detected. While we observe elevated TNFSF10 levels in cells with globular aggregates, its accumulation may reflect impaired secretion rather than necessarily indicating imminent cell death.

Based on the reviewer's feedback and our data showing sequential phenotypic changes culminating in this distinctive state, we have revised our terminology to describe this as a 'late-stage' hepatocyte phenotype. This better reflects our observations of a distinct proteomic signature characterized by globular aggregates, elevated TNFSF10, and altered cellular functions. In addition, we toned down the apoptosis statements, for instance by removing the hypothesis that the late-stage phenotype precedes "intrinsic or extrinsic apoptosis" (l. 230).

It is likely that larger complexes, such as CRP heximers, might become trapped by the solidification of Z-AAT that has recently been described in the ER. Do the current data suggest any bias in the nature of ER client proteins that accumulate in cells in this 'advanced' state, for example are members of larger complexes less likely to be secreted?

While we agree with the hypothesis raised by the reviewer, our data does not suggest a bias in the nature of ER client proteins that accumulate in AAT+ cells. We investigated this question by retrieving information about protein complexes from two databases, CORUM and HuMAP 2.0, and annotated all proteins in our DVP experiment as complex-bound or not. This did not reveal striking differences.

Figure R4, Potential overrepresentation of protein complexes in AAT+ cells. Complexes were annotated with two databases.

References: appropriate credit to previous work?

The referencing appears appropriate.

Clarity and context: lucidity of abstract/summary, appropriateness of abstract, introduction and conclusions

These sections of the manuscript are very well written.

We thank reviewer #3 for their time to critically read and comment on the manuscript.

References

1. Chambers, J. E. *et al.* Z- α 1-antitrypsin polymers impose molecular filtration in the endoplasmic reticulum after undergoing phase transition to a solid state. *Sci. Adv.* **8**, eabm2094 (2022).
2. Zhang, Y. *et al.* LMAN1-MCFD2 complex is a cargo receptor for the ER-Golgi transport of α 1-antitrypsin. *Biochem. J.* **479**, 839–855 (2022).
3. Whisstock, J. C. & Bottomley, S. P. Molecular gymnastics: serpin structure, folding and misfolding. *Curr. Opin. Struct. Biol.* **16**, 761–768 (2006).

4. Spivak, I. *et al.* Alpha-1 Antitrypsin Inclusions Sequester GRP78 in a Bile Acid-Inducible Manner. *Liver Int.* **45**, e16207 (2025).
5. Clark, V. C. *et al.* Clinical and histologic features of adults with alpha-1 antitrypsin deficiency in a non-cirrhotic cohort. *J. Hepatol.* **69**, 1357–1364 (2018).
6. Niu, L. *et al.* Noninvasive proteomic biomarkers for alcohol-related liver disease. *Nat. Med.* **28**, 1277–1287 (2022).
7. Mund, A. *et al.* Deep Visual Proteomics defines single-cell identity and heterogeneity. *Nat. Biotechnol.* **40**, 1231–1240 (2022).
8. Nordmann, T. M. *et al.* A Standardized and Reproducible Workflow for Membrane Glass Slides in Routine Histology and Spatial Proteomics. *Mol. Cell. Proteomics MCP* **22**, 100643 (2023).
9. Rosenberger, F. A. *et al.* Spatial single-cell mass spectrometry defines zonation of the hepatocyte proteome. *Nat. Methods* **20**, 1530–1536 (2023).
10. Fan, R. Integrative spatial protein profiling with multi-omics. *Nat. Methods* **21**, 2223–2225 (2024).
11. Brzowska, N. *et al.* Convergent evolution of somatic escape variants in SERPINA1 in the liver in alpha-1 anti-trypsin deficiency. 2024.09.02.610686 Preprint at <https://doi.org/10.1101/2024.09.02.610686> (2024).